# FAM83D directs protein kinase CK1α to the mitotic spindle for proper spindle positioning

Luke J Fulcher[1], Zhengcheng He[2] (iD), Lin Mei[2], Thomas J Macartney[1], Nicola T Wood[1], Alan R Prescott[3], Arlene J Whigham[4], Joby Varghese[1], Robert Gourlay[1], Graeme Ball[3], Rosemary Clarke[4], David G Campbell[1], Christopher A Maxwell[2] & Gopal P Sapkota[1,*] (iD)

## Abstract

The concerted action of many protein kinases helps orchestrate the error-free progression through mitosis of mammalian cells. The roles and regulation of some prominent mitotic kinases, such as cyclin-dependent kinases, are well established. However, these and other known mitotic kinases alone cannot account for the extent of protein phosphorylation that has been reported during mammalian mitosis. Here we demonstrate that CK1α, of the casein kinase 1 family of protein kinases, localises to the spindle and is required for proper spindle positioning and timely cell division. CK1α is recruited to the spindle by FAM83D, and cells devoid of *FAM83D*, or those harbouring CK1α-binding-deficient *FAM83D^F283A/F283A* knockin mutations, display pronounced spindle positioning defects, and a prolonged mitosis. Restoring *FAM83D* at the endogenous locus in *FAM83D^−/−* cells, or artificially delivering CK1α to the spindle in *FAM83D^F283A/F283A* cells, rescues these defects. These findings implicate CK1α as new mitotic kinase that orchestrates the kinetics and orientation of cell division.

**Keywords** CK1; FAM83D; kinase; mitosis; spindle positioning
**Subject Category** Cell Cycle

## Introduction

The prominent mitotic roles for kinases such as cyclin-dependent kinases (CDKs), Aurora kinases, Polo-like kinases (PLKs) and Nima-related kinases (NEKs) have been well characterised [1–7]. However, the role of Casein Kinase 1 alpha (CK1α) in mitosis, if any, remains poorly defined. CK1α belongs to the CK1 family of Ser/Thr protein kinases that are implicated in diverse roles in a whole plethora of cellular processes, from Wnt signalling to the regulation of circadian rhythms [8]. CK1 isoforms can phosphorylate hundreds of proteins *in vitro*, with a preference for Ser/Thr residues that conform to either a D/E-X-X-S*/T* or pS/pT-X-X-S*/T* motif [9]. A recent mitotic phosphoproteomic study found that around half of the identified phosphorylation sites conformed to the predicted CK1-consensus phosphorylation motifs [10], potentially implying a significant role for CK1 catalytic activity in mitotic protein phosphorylation. Yet, there is a lack of definitive evidence regarding whether and how any of the CK1 isoforms, or CK1α in particular, are involved in mitosis.

Regarded as constitutively active protein kinases, the regulation of CK1 isoforms is critically important, yet poorly understood, especially when considering their participation in multiple, diverse, cellular functions in many, different cellular compartments [8,11]. We recently reported that the FAM83 family of poorly characterised proteins act as subcellular anchors for CK1 isoforms through the conserved N-terminal domain of unknown function 1669 (DUF1669) [12]. Our findings that FAM83 proteins interact and co-localise with different CK1 isoforms offer the tantalising possibility that FAM83 proteins direct CK1 isoforms to specific subcellular compartments, and in doing so, regulate their substrate availability/accessibility [13]. In line with this, we have shown that FAM83G [(aka protein associated with SMAD1 (PAWS1)] activates Wnt signalling through its association with CK1α [14]. Here, we sought to investigate the FAM83D protein, and define its physiological role in relation to CK1 isoforms. Although FAM83D (aka spindle protein CHICA) is poorly characterised, it has been shown to be recruited to the mitotic spindle through its association with the microtubule-associated protein hyaluronan-mediated motility receptor (HMMR, aka RHAMM or CD168) [15–17]. Unlike the other FAM83 members that appear to associate robustly with CK1α, we found that over-expressed green fluorescent protein (GFP)-tagged FAM83D in asynchronous cell extracts interacted rather weakly, yet selectively, with CK1α [12], suggesting this could be a regulated interaction. Consistent with this, in the course of an unbiased proteomic approach to identify interactors of endogenous FAM83D from both asynchronous and mitotic extracts, we discovered in this study that FAM83D interacts with CK1α only in mitosis. As the kinetics of

1 Medical Research Council, Protein Phosphorylation and Ubiquitylation Unit, University of Dundee, Dundee, UK
2 Michael Cuccione Childhood Cancer Program, British Columbia Children's Hospital, University of British Columbia, Vancouver, BC, Canada
3 Dundee Imaging Facility, School of Life Sciences, University of Dundee, Dundee, UK
4 Flow Cytometry and Sorting Facility, School of Life Sciences, University of Dundee, Dundee, UK
*Corresponding author. Tel: +44 1382 386330; E-mail: g.sapkota@dundee.ac.uk

chromosomal alignment are delayed and the cell division axis is altered following the depletion of *FAM83D* by siRNA, as well as in cells lacking *HMMR* or those derived from *HMMR* knockout mice [15–17], we hypothesised that these phenotypes could be potentially explained by the non-delivery of CK1α to the spindle in the absence of FAM83D or HMMR. Here, we show that the FAM83D–CK1α interaction is critically important for correct and efficient spindle positioning, as well as smooth progression through the cell division cycle.

# Results

### FAM83D and CK1α interact only in mitosis

In order to investigate the role of FAM83D at physiological levels, we first generated a *FAM83D* knockout U2OS cell line (*FAM83D*$^{−/−}$) (Fig EV1), along with a U2OS cell line harbouring an in-frame homozygous knockin of a GFP tag at the C-terminus of the endogenous *FAM83D* gene (*FAM83D*$^{GFP/GFP}$) (Fig EV1), with CRISPR/Cas9 gene editing technology, and verified these by Western blotting (Fig 1A) and DNA sequencing. Given the links between FAM83D and mitosis [16,17], we next undertook an unbiased proteomic approach to identify interactors of endogenous FAM83D-GFP from either asynchronous or mitotic *FAM83D*$^{GFP/GFP}$ knockin cell extracts. Mitotic cells were collected by shake-off, following either prometaphase arrest with nocodazole and a brief release into fresh medium to allow them to progress into mitosis, or mitotic arrest with the Eg5 chromokinesin inhibitor S-trityl L-cysteine (STLC), which results in monopolar spindle formation [18]. Mass spectrometric analysis of anti-GFP immunoprecipitates (IPs) from both asynchronous and mitotic cell extracts identified several known FAM83D interactors, including HMMR, dynein light chain 1 (DYNLL1) and the transcription factor BTB domain and CNC homolog 1 (BACH1) [12,16,19] (Fig 1B), potentially revealing the constitutive FAM83D interactors. Excitingly, the only interactor of FAM83D that was robustly identified from mitotic, but not asynchronous extracts, was CK1α (Fig 1B). The mitotic interactions observed between FAM83D and CK1α or BACH1 constitute novel findings (Fig 1C).

We sought to validate the interaction between FAM83D and CK1α at the endogenous level. Endogenous CK1α was detected in anti-GFP IPs from nocodazole-synchronised mitotic but not asynchronous *FAM83D*$^{GFP/GFP}$ cell extracts (Fig 1D). CK1δ and ε did not interact with FAM83D, but the known FAM83D interactors HMMR and DYNLL1 were detected in FAM83D IPs from both asynchronous and mitotic cell extracts, suggesting a constitutive mode of interaction (Fig 1D). A striking electrophoretic mobility shift of FAM83D was observed in mitosis compared to asynchronous cells, suggesting a potential post-translational modification (PTM) (Fig 1D). In mitotic extracts collected from *FAM83D*$^{GFP/GFP}$ cells following synchronisation with STLC, endogenous CK1α was also detected in anti-GFP IPs (Fig 1E). Moreover, in cells that were exposed to either nocodazole or STLC but were non-mitotic (i.e. adherent cells that did not shake-off following drug treatment), no interaction between CK1α and FAM83D was observed (Fig 1E), ruling out possible drug-dependent stimulation of the CK1α–FAM83D interaction. To rule out the possibility that the GFP tag on FAM83D might influence its interaction with CK1α, we employed the 2G *PAI1* U2OS cell line,

which harbours a non-fused GFP tag on the *PAI1* locus [20], as a control. Indeed, anti-GFP IPs from these cells did not co-precipitate CK1α (Fig 1E). Cell cycle stages in asynchronous and mitotic cells were confirmed by flow cytometry following propidium iodide staining (Fig 1F). At the endogenous level in wild-type U2OS cells, CK1α was detected in anti-FAM83D IPs only from mitotic but not asynchronous extracts, while FAM83D was identified in both. Neither FAM83D nor CK1α were detected in control IgG IPs (Fig 1G). The mitotic electrophoretic mobility shift of endogenous FAM83D was also apparent in wild-type U2OS cells (Fig 1G). Endogenous FAM83D was detected in anti-CK1α IPs only from mitotic but not asynchronous extracts, while CK1α was identified in both (Fig 1H).

Next, to decipher exactly when FAM83D associates with CK1α during the cell division cycle, we arrested *FAM83D*$^{GFP/GFP}$ cells in G2 using the CDK1 inhibitor RO-3306 [21], or in mitosis (M) with STLC shake-off, and lysed at 0 (M), 2 (M), 4 (G1) and 6 (G1) hours after STLC washout (Fig 1I). Cell cycle stages were assigned by monitoring the levels of cyclin B1 (high in M) and cyclin A2 (high in G2), as well as flow cytometry (Fig 1I and J). CK1α was only detected in anti-GFP IPs from the mitotic extracts, but not from G2-arrested, G1 or asynchronous extracts (Fig 1I). As observed before, FAM83D-GFP displayed a robust mobility shift only in the mitotic samples (Fig 1I). To test whether the mitotic FAM83D–CK1α interaction occurs in other mammalian cells, we expanded our interaction analysis to include three additional human cell lines, namely cervical cancer HeLa, lung adenocarcinoma A549 and human keratinocyte HaCaT. Endogenous FAM83D protein immunoprecipitated from these cells in STLC-synchronised mitotic extracts also co-precipitated CK1α (Fig 1K). As in U2OS cells, FAM83D also displayed a pronounced electrophoretic mobility shift in mitotic extracts isolated from these cell lines (Fig 1K).

We sought to probe the mitosis-dependent nature of the FAM83D–CK1α interaction further. Given that the N-terminal DUF1669 of FAM83 proteins mediates CK1α binding [12], we reasoned that the C-terminus of FAM83D may act in an auto-inhibitory manner to prevent CK1α binding until mitosis. In agreement with this hypothesis, isolated FAM83D C-terminal fragments lacking the DUF1669 co-precipitate N-terminal FAM83D fragments, yet do not co-precipitate CK1α (Fig EV2A). In contrast, the N-terminal DUF1669-containing fragment of FAM83D co-precipitated CK1α even under asynchronous conditions (Fig EV2B), suggesting that the DUF1669 of FAM83D is both necessary and sufficient to interact with CK1α, and that during mitosis the autoinhibitory effect of the FAM83D C-terminal fragment on the DUF1669 is relieved.

### FAM83D recruits CK1α to the mitotic spindle

We sought to test whether FAM83D and CK1α interact and co-localise in cells during mitosis. In mitotic *FAM83D*$^{GFP/GFP}$ cells, we observed complete overlapping signals between the FAM83D-GFP fluorescence and endogenous CK1α immunofluorescence (IF) signals on the STLC-induced monopolar mitotic spindles (Fig 2A, top panel). In wild-type cells, which express *FAM83D* without the GFP tag, we also observed CK1α on mitotic spindles (Fig 2A, middle panel). Strikingly, consistent with our hypothesis that FAM83 proteins recruit CK1 isoforms to distinct cellular sites, no CK1α signal was evident on the spindle apparatus in *FAM83D*$^{−/−}$

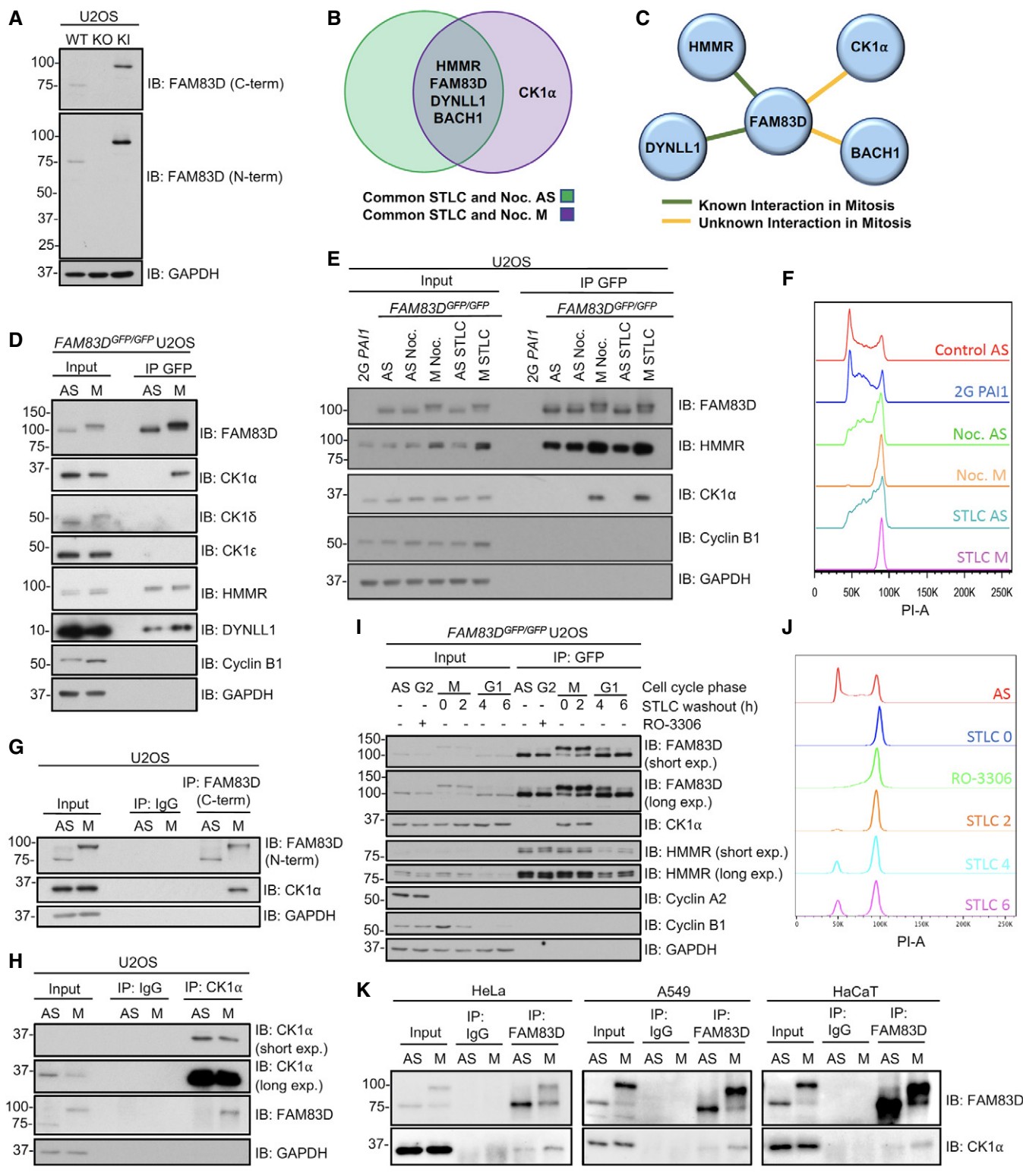

Figure 1.

U2OS cells (Fig 2A; lower panel), suggesting FAM83D recruits CK1α to the spindle in mitosis. No CK1ε staining was detected at the spindle apparatus in either the wild-type, *FAM83D*⁻/⁻ or *FAM83D*ᴳᶠᴾ/ᴳᶠᴾ cells (Fig EV2C), highlighting the specificity of the

FAM83D–CK1α interaction. HMMR, which is responsible for recruiting FAM83D to the spindle in mitosis [16], was observed at the spindle in wild-type, *FAM83D*⁻/⁻ and *FAM83D*ᴳᶠᴾ/ᴳᶠᴾ cells (Fig EV2D).

◄

**Figure 1.  FAM83D and CK1α interact only in mitosis.**

A       Immunoblot analysis of wild-type (WT), *FAM83D*$^{-/-}$ knockout (KO) and *FAM83D*$^{GFP/GFP}$ knockin (KI) U2OS cell lines.
B       Proteomic analysis on asynchronous (AS), nocodazole- or STLC-synchronised mitotic (M) *FAM83D*$^{GFP/GFP}$ knockin (KI) U2OS cells. The Venn diagram depicts the top proteins which were identified as FAM83D interactors in AS, M or both AS and M conditions, in both nocodazole and STLC treatments (for a detailed analysis procedure, see the Materials and Methods section).
C       Schematic highlighting whether a mitotic interaction was previously known between FAM83D and the interacting proteins identified in (B).
D       AS or nocodazole-synchronised M KI cells were lysed and subjected to GFP TRAP immunoprecipitations (IP). Extracts (input) and IP samples were analysed by immunoblotting (IB) with the indicated antibodies.
E       KI cells synchronised in mitosis with either nocodazole (M Noc.) or STLC (M STLC) were collected by shake-off, and drug-treated cells that remained adherent after shake-off (AS Noc.; AS STLC) were lysed and subjected to GFP TRAP IP. AS cells and free GFP-expressing 2G-*PAI1* U2OS cells were included as controls. Input and IP samples were analysed by IB with the indicated antibodies.
F       Propidium iodide staining analyses revealing cell cycle distribution profiles for the samples described in (E).
G, H    AS or nocodazole-synchronised (M) WT U2OS cells were subjected to IP with IgG and either anti-FAM83D-coupled sepharose beads (G), or anti-CK1α-coupled sepharose beads (H). Input and IP samples were analysed by IB with the indicated antibodies.
I       KI cells were synchronised in G2 with RO-3306, or arrested in mitosis (M) using STLC. STLC-treated shake-off cells were washed and re-plated, and cells lysed at the indicated time points after STLC washout. Cell lysates were subjected to GFP TRAP IP and input and IP extracts analysed by IB with the indicated antibodies.
J       Propidium iodide staining analyses revealing cell cycle distribution profiles for the samples described in (I).
K       AS or STLC-synchronised (M) HeLa, A549 and HaCaT cells were subjected to IP with either IgG- or anti-FAM83D-coupled sepharose beads. Input and IP samples were analysed by IB with the indicated antibodies.

Data information: All blots are representative of at least three independent experiments.
Source data are available online for this figure.

We previously identified two conserved residues (equivalent to D$^{249}$ and F$^{283}$ of FAM83D) within the conserved DUF1669 of FAM83 proteins that were critical for mediating the FAM83–CK1 interaction [12]. GFP-tagged wild-type *FAM83D*, but not F283A or D249A mutants, co-precipitated endogenous CK1α during mitosis when transiently expressed in *FAM83D*$^{-/-}$ cells (Fig EV3A). Interestingly, we noted that the mitotic electrophoretic shift evident for wild-type GFP-FAM83D was absent for the two mutants (Fig EV3A), suggesting that CK1α binding is required for the mitotic mobility shift evident in FAM83D. To validate these findings at the endogenous level, we generated U2OS knockin cell lines harbouring the F283A mutation on *FAM83D* as well as a GFP tag at the C-terminus (hereafter referred to *FAM83D*$^{GFP/GFP(F283A)}$, Fig EV1) using CRISPR/Cas9, and verified homozygous knockins by DNA sequencing. Excitingly, like *FAM83D*$^{-/-}$ cells, no endogenous CK1α IF signal was detected on the spindles in these cells, while overlapping GFP and CK1α signals were observed in *FAM83D*$^{GFP/GFP}$ cells (Fig 2B). However, FAM83D(F283A)-GFP still localised to the mitotic spindle, albeit with relatively less GFP fluorescence intensity compared to FAM83D-GFP (Fig 2B). Quantification of the CK1α IF signal on the mitotic spindle from these cells corroborated these observations (Fig 2C). Furthermore, endogenous CK1α was detected only in GFP IPs from mitotic *FAM83D*$^{GFP/GFP}$ cell extracts, but not from mitotic *FAM83D*$^{-/-}$ or *FAM83D*$^{GFP/GFP(F283A)}$ extracts (Fig 2D). Identical results were obtained with an independent CRISPR *FAM83D*$^{GFP/GFP(F283A)}$ knockin clone, further confirming these results (Fig EV3B and C). To further ascertain whether CK1α recruitment to the spindle was dependent on the FAM83D protein, we employed the Affinity-directed PROtein Missile (AdPROM) system [22,23] to efficiently degrade FAM83D-GFP using VHL fused to an anti-GFP nanobody (VHL-aGFP.16) (Figs 2E and F, and EV1). In mitotic *FAM83D*$^{GFP/GFP}$ cells, and *FAM83D*$^{GFP/GFP}$ cells expressing the VHL or aGFP.16 controls, overlapping GFP fluorescence and endogenous CK1α IF signals were evident. In contrast, both signals disappeared from mitotic spindles in mitotic *FAM83D*$^{GFP/GFP}$ cells expressing the VHL-aGFP.16 AdPROM, which resulted in FAM83D-GFP degradation (Fig 2F and G).

Finally, we rescued the *FAM83D*$^{-/-}$ U2OS cells by knocking in a polycistronic cassette, consisting of wild-type *FAM83D* cDNA, an IRES element, *GFP* reporter and polyadenosine tail, directly downstream of the endogenous *FAM83D* promoter (Figs 3A and EV1) and obtained and verified two rescue clones by DNA sequencing and immunoblotting (Fig 3B). In both clones, we verified that the expression of the FAM83D protein and its mitotic phospho-mobility shift closely mirrored that seen in wild-type cells; however, the expression of *FAM83D* in clone 11 was slightly lower than that observed in clone 6 (Fig 3B). Next, we verified that both FAM83D (Fig 3C) and CK1α (Fig 3D and E) localised to the spindle apparatus in both *FAM83D*-rescue clones, similar to that observed in wild-type cells. Finally, we also confirmed that the reinstated *FAM83D* in both clones co-precipitated endogenous CK1α only in mitosis (Fig 3F).

## FAM83D and CK1α in unperturbed cells

In order to determine endogenous FAM83D and CK1α co-localisation at different phases of mitosis in the absence of any drug-induced cell synchronisation, we first knocked in an mCherry tag onto either the *CSNK1A1* gene (CK1α) or the *CSNK1E* gene (CK1ε) in *FAM83D*$^{GFP/GFP}$ U2OS cells by CRISPR/Cas9 (Fig EV1), and verified homozygous insertions by immunoblotting and IPs (Fig EV4A and B), as well as genomic DNA sequencing. After confirming that the mCherry tag did not render CK1α inactive (Fig EV4C), we performed fluorescence microscopy on these cells. We observed robust overlapping centrosomal and mitotic spindle fluorescence between FAM83D-GFP and mCherry-CK1α from prometaphase all the way to anaphase, with less intense co-fluorescence evident in the latter stages of mitosis (Fig 3G). In contrast, we did not observe spindle localisation of mCherry-CK1ε, nor co-localisation with FAM83D-GFP (Fig EV4D), at any stage of mitosis.

## FAM83D regulation during the cell cycle

The ~25 kDa electrophoretic mobility shift evident for mitotic FAM83D-GFP collapsed substantially when the FAM83D-GFP IPs

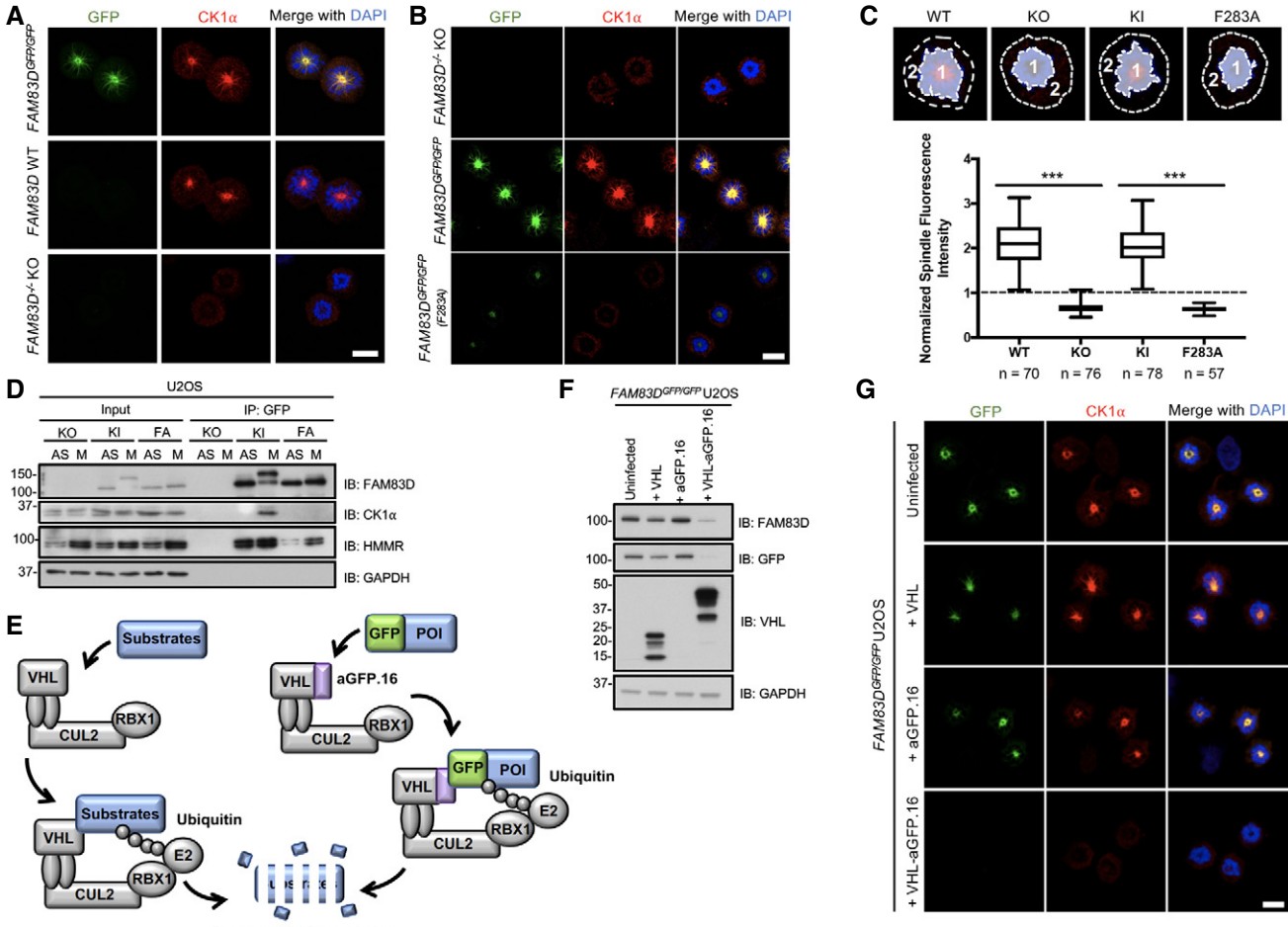

**Figure 2. FAM83D recruits CK1α to the spindle.**

A STLC-synchronised mitotic (M) wild-type (WT), *FAM83D*[−/−] knockout (KO) and *FAM83D*[GFP/GFP] knockin (KI) U2OS cells were subjected to anti-CK1α immunofluorescence and GFP fluorescence microscopy. DNA is stained with DAPI. Scale bars, 20 μm.

B STLC-synchronised mitotic (M) *FAM83D*[−/−] knockout (KO), *FAM83D*[GFP/GFP] knockin (KI) and *FAM83D*[GFP/GFP(F283A)] knockin (FA) U2OS cells were subjected to anti-CK1α immunofluorescence and GFP fluorescence microscopy. DNA is stained with DAPI. Scale bars, 20 μm.

C Quantification of CK1α spindle localisation for the cells described in panels (A and B). Cell images denote the measured regions used to calculate the ratios on the box plot. Box plot whiskers denote the minimum and maximum measured values. The middle line represents the median, and the box ranges depict the 25th/75th percentiles. ***$P < 0.0001$; ANOVA. Analysis was performed on the indicated number of cells, $n = 2$.

D The cell lines described in (B) were STLC-synchronised and mitotic cells (M) isolated by shake-off. Asynchronous (AS) cells were included as a control. Cells were lysed and subjected to GFP TRAP immunoprecipitation (IP) and subsequent immunoblotting (IB) with the indicated antibodies.

E Schematic illustration of the AdPROM-mediated degradation of FAM83D. VHL; Von Hippel-Lindau protein, CUL2; cullin 2, RBX1; RING-box protein 1, E2; E2 ubiquitin-conjugating enzyme, aGFP.16; anti-GFP.16 nanobody.

F KI cells were infected with retroviruses encoding either VHL, aGFP.16 or VHL-aGFP.16. Uninfected cells were used as a control. Cells were lysed and subjected to IB with the indicated antibodies.

G The cell lines described in (F) were subjected to anti-CK1α immunofluorescence and GFP fluorescence microscopy. DNA is stained with DAPI. Scale bars, 20 μm.

Data information: All blots are representative of at least three independent experiments.
Source data are available online for this figure.

were subjected to λ-phosphatase treatment, almost to the level of FAM83D-GFP in asynchronous cells (Fig 4A), suggesting that this mobility shift was due to phosphorylation. Using the mobility shift as a readout, we investigated FAM83D phosphorylation over the course of a cell division following release from STLC arrest. As expected, there was a reduction in levels of cyclin B1 and phospho-histone H3 phosphorylation as cells progressed through mitosis [24,25], indicating that cells started exiting mitosis around 2 h after

STLC washout (Fig 4B). Interestingly, concurrent reduction in the levels of both HMMR and phospho-FAM83D was observed, suggesting that both proteins are regulated in a cell cycle-dependent manner (Fig 4B). Such patterns in the levels of cell cycle-regulated proteins are often associated with their degradation following mitotic exit, or during the metaphase-to-anaphase transition [25,26]. Interestingly, the protein levels of CK1α did not change after the STLC washout (Fig 4B). Considering that both FAM83D and CK1α

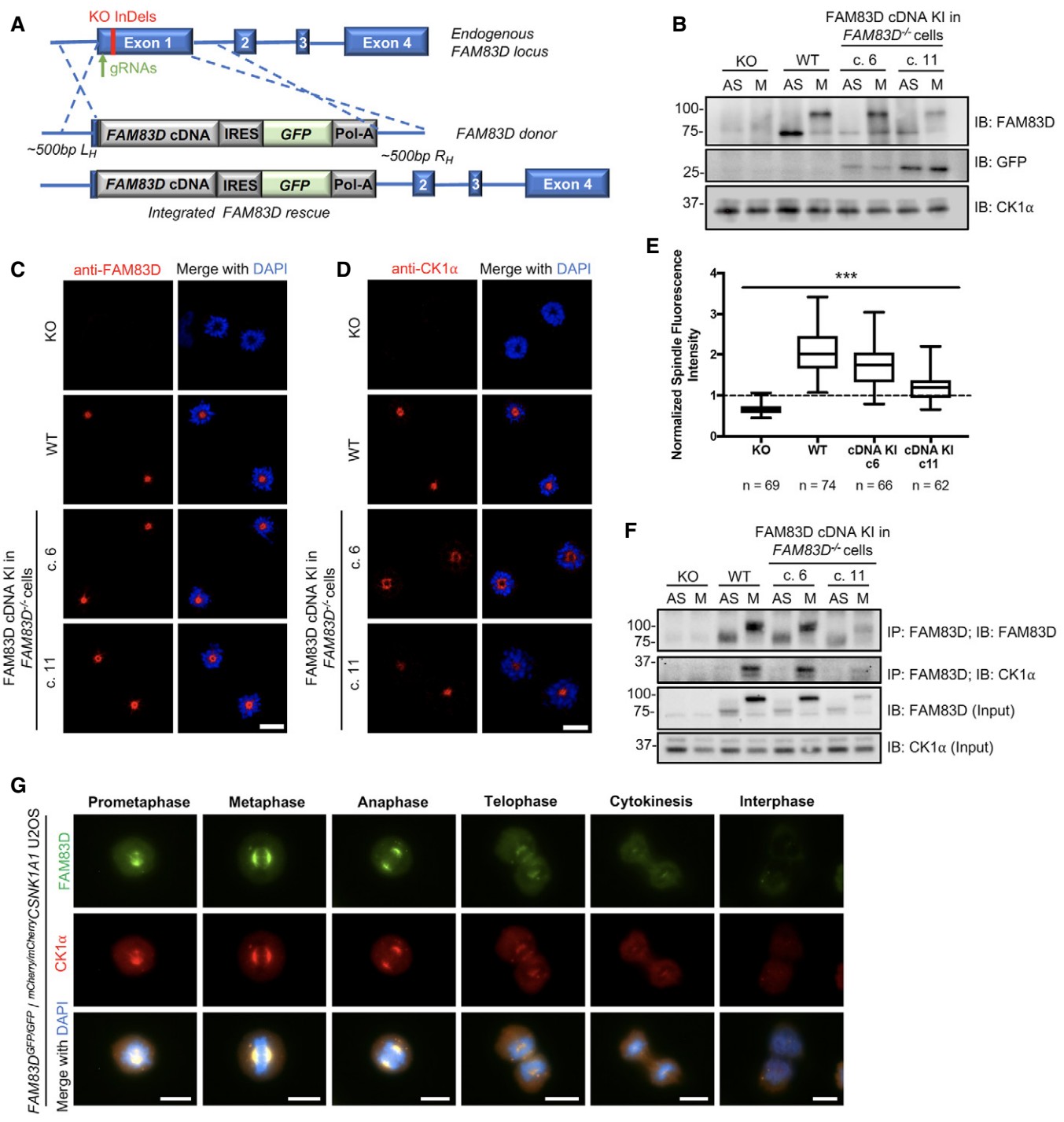

**Figure 3.**

appear to dissociate from the spindle following the metaphase-to-anaphase transition (Fig 3G), it appears likely that following FAM83D degradation, CK1α can no longer localise to the spindle, and dissociates into the cytosol. The reduction in FAM83D and HMMR protein levels following STLC washout was blocked with MG132 (Fig 4C), suggesting that FAM83D and HMMR potentially undergo proteasomal degradation upon mitotic exit. Cyclin B1 levels were also rescued by MG132 treatment (Fig 4C). As MG132 is also

known to inhibit the metaphase-to-anaphase transition through the stabilisation of the anaphase-promoting complex/cyclosome (APC/C) E3 ligase substrate securin [25], this transitional delay could in turn explain the lack of phospho-FAM83D and HMMR degradation. Indeed, analogous results on phospho-FAM83D and HMMR stabilisation were observed when mitotic cells were released into medium containing the APC/C inhibitor ProTAME [27] (Fig 4D). The CK1α protein levels were unaffected by either MG132 or ProTAME

**Figure 3. Reinstating *FAM83D* at the endogenous locus in *FAM83D* knockout cells.**

A    Schematic illustrating the CRISPR-based strategy used to reintroduce FAM83D into the *FAM83D$^{-/-}$* knockout background.
B    *FAM83D$^{-/-}$* (KO), wild-type (WT) and two independent clones from a CRISPR/Cas9-mediated knockin rescue of *FAM83D* cDNA into *FAM83D$^{-/-}$* cells (c. 6 and c. 11) were synchronised in mitosis (M) with STLC. Asynchronous (AS) cells were included as a control. Cells were lysed and subjected to immunoblotting (IB) with the indicated antibodies.
C, D  The cell lines described in (B) were STLC-synchronised in mitosis, fixed and stained with antibodies recognising FAM83D (C) or CK1α (D). Representative images of mitotic cells are included. Scale bars, 20 μm.
E    Quantification of CK1α spindle localisation for the experiment described in (D) using the same strategy employed in Fig 2C. Box plot whiskers denote the minimum and maximum measured values. The middle line represents the median, and the box ranges depict the 25$^{th}$/75$^{th}$ percentiles. ***$P$ < 0.0001; ANOVA. Analysis was performed on the indicated number of cells, $n$ = 1.
F    The cell lines described in (B) were STLC-synchronised in mitosis (M) or left AS, lysed and subjected to immunoprecipitation (IP) with anti-FAM83D-coupled sepharose beads, before IB with the indicated antibodies.
G    Asynchronous (AS) *FAM83D$^{GFP/GFP}$/$^{mCherry/mCherry}$CSNK1A1* knockin U2OS cells were fixed and imaged. Representative images from the indicated cell cycle stages are included. Scale bars, 10 μm. All blots are representative of at least three independent experiments.

Data information: All blots are representative of at least three independent experiments.
Source data are available online for this figure.

treatments (Fig 4C and D). Next, we sought to confirm that the mitotic post-translational machinery triggers FAM83D degradation in cells. First, we arrested U2OS cells in mitosis with STLC and isolated them by shake-off. Mitotic cells were released into medium containing either the CDK1 inhibitor RO-3306 or dimethyl sulfoxide (DMSO) vehicle control, in the presence or absence of the proteasomal inhibitor MG132. Forced mitotic exit caused by CDK1 inhibition resulted in robust FAM83D, HMMR and cyclin B1 degradation, whereas addition of MG132 blocked their proteolysis (Fig 4E), thereby confirming that it is the mitotic post-translational machinery that regulates FAM83D stability as cells progress through mitosis. Transcriptional analysis by qRT–PCR showed a significant twofold increase in *FAM83D* and *HMMR* transcript levels in mitotic over asynchronous cells (Fig 4F), suggesting that both *FAM83D* and *HMMR* are cell cycle-regulated genes, similar to *CCNB1* transcripts [28] (Fig 4F). We did not detect any significant difference in CK1α transcript levels between asynchronous and mitotic cells (Fig 4F).

The synergistic regulation of FAM83D and HMMR, along with their constitutive interaction, suggested a possible role for HMMR in the regulation of FAM83D and CK1α in mitosis (Fig 4G). Indeed, as observed in U2OS cells, a robust mitotic phospho-FAM83D mobility shift was evident in wild-type MEFs, but this shift was completely absent in the *HMMR* knockout MEFs [15] (Fig 4H). This is consistent with the notion that HMMR directs FAM83D to the spindle [16], and hence, in the absence of HMMR, FAM83D no longer localises to the mitotic spindle and is not phosphorylated. If this were the case, one would expect that CK1α should not be recruited to the mitotic spindle in *HMMR* knockout MEFs. After first confirming HMMR localises to the spindle in the wild-type but not *HMMR* knockout MEFs (Fig 4I), we observed robust CK1α mitotic spindle localisation in wild-type MEFs, but could not detect CK1α on the mitotic spindle in *HMMR* knockout MEFs (Fig 4J). Collectively, these data support the model where HMMR directs FAM83D to the spindle apparatus in mitosis, and subsequently, FAM83D recruits CK1α through the DUF1669. Furthermore, these findings show that the mitotic phospho-dependent mobility shift observed for FAM83D relies on the recruitment of CK1α to the mitotic spindle.

**The role of CK1α in FAM83D phosphorylation**

Given the absence of the FAM83D(F283A)-GFP mitotic mobility shift despite its mitotic localisation (Fig 2B and D), we wondered whether targeted delivery of CK1α to FAM83D(F283A)-GFP by an anti-GFP nanobody (aGFP.16) could artificially reconstitute the FAM83D–CK1α interaction and restore the phospho-shift (Figs 5A and EV1). Excitingly, expression of aGFP.16-CK1α in *FAM83D$^{GFP/GFP(F283A)}$* cells rescued the phospho-mobility shift of FAM83D(F283A)-GFP, regardless of whether the cells were in mitosis or not (Fig 5B). We also observed robust association of aGFP.16-CK1α with FAM83D(F283A)-GFP in anti-GFP IPs (Fig 5B). During the course of these experiments, we noticed that this rescued FAM83D(F283A)-GFP mobility shift was slightly smaller than the mitotic mobility shift of wild-type FAM83D-GFP (Fig 5B). Thus, to confirm that aGFP.16-CK1α was not phosphorylating random residues on FAM83D, we repeated this experiment in wild-type *FAM83D$^{GFP/GFP}$* U2OS cells (Fig EV5A). As with FAM83D(F283A)-GFP, we observed constitutive FAM83D-GFP phosphorylation regardless of cell cycle stage, but the aGFP.16-CK1α-induced band-shift migrated to the same level as the mitotic FAM83D-GFP shift in control cells (Fig EV5A). As we do not observe any larger mitotic mobility shifts when wild-type FAM83D is phosphorylated by aGFP.16-CK1α, we attribute the slight reduction in mobility seen with the aGFP.16-CK1α-rescued FAM83D(F283A)-GFP mutant to potential conformational defects resulting from the F283A mutation, potentially occluding one or more of the phospho-acceptor residues from aGFP.16-CK1α.

The mobility shift of FAM83D(F283A)-GFP restored by the targeted delivery of CK1α relied on CK1α catalytic activity as two catalytically inactive mutants of CK1α (K46D and D136N) failed to rescue this phospho-shift completely (Fig 5C). Critically, we observed robust co-localisation between FAM83D and CK1α on intact mitotic spindles (Fig 5D). However, under conditions where CK1α phosphorylated FAM83G *in vitro*, it failed to phosphorylate recombinant FAM83D (Fig EV5B), suggesting a cellular FAM83D context or a priming phosphorylation event might be required for FAM83D phosphorylation by CK1α. Interestingly, when wild-type CK1ε, which does not interact with FAM83D nor localise to the spindle, is delivered to FAM83D(F283A)-GFP by aGFP.16, it too rescued the FAM83D phospho-shift (Fig EV5C), suggesting the proximal catalytic activity of CK1 is sufficient for the phospho-shift and highlights the notion that subcellular localisation and substrate association are important determinants for CK1 targets.

By delivering CK1α to the mitotic spindle, it is likely that FAM83D directs CK1α to phosphorylate many mitotic substrates, including itself. However, this hypothesis relies on CK1α being

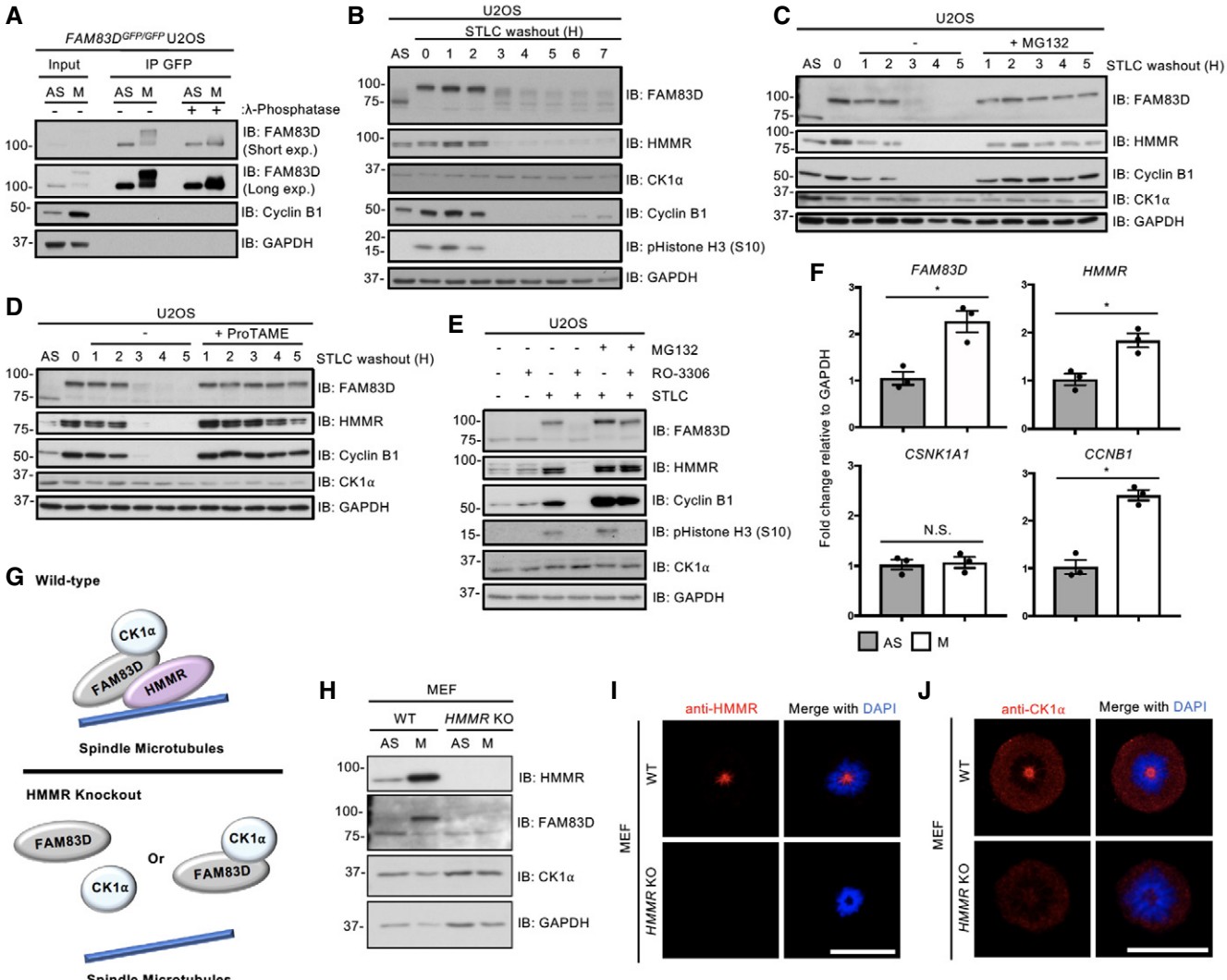

**Figure 4. FAM83D regulation during the cell cycle.**

A     Nocodazole-synchronised mitotic *FAM83D*^GFP/GFP^ knockin U2OS cells were lysed and subjected to GFP TRAP immunoprecipitation (IP), followed by incubation ± λ-phosphatase. Asynchronous (AS) cells were used as a control. Whole-cell extracts (input) and IP samples were subjected to SDS–PAGE and subsequent immunoblotting (IB) with the indicated antibodies.

B–D   STLC-synchronised mitotic wild-type U2OS cells were lysed at the indicated time points following STLC washout and release into medium (B), medium ± MG132 (C) or medium ± ProTAME (D). AS cells were used as a control. Lysed extracts were subjected to IB with the indicated antibodies.

E     Wild-type U2OS cells were either left asynchronous (AS), or arrested in mitosis with STLC and collected by shake-off (M). AS and M cells were incubated in media containing combinations of RO-3306 and MG132 as indicated, prior to lysis. MG132 was applied for 1.5 h, whereas RO-3306 was applied for the last 1 h of incubation prior to lysis. Samples were lysed and extracts were subjected to SDS–PAGE, before IB with the indicated antibodies.

F     STLC-synchronised mitotic wild-type U2OS cells were subjected to qRT–PCR analysis using primers for *FAM83D*, *HMMR*, *CSNK1A1* and *CCNB1*. AS cells were used as a control. Error bars, SEM; *P < 0.01; Student's *t*-test. n = 3.

G     Schematic representing the predicted effects of *HMMR* knockout on FAM83D–CK1α delivery to the mitotic spindle. In the absence of HMMR, no FAM83D and, by extension, no CK1α can localise to the spindle.

H     STLC-synchronised mitotic wild-type (WT) and *HMMR* knockout (KO) mouse embryonic fibroblasts (MEFs) were lysed and subjected to IB with the indicated antibodies. AS cells were included as a control.

I, J   The cells described in (H) were STLC-synchronised in mitosis and subjected to immunofluorescence microscopy with an anti-HMMR antibody (I) or an anti-CK1α antibody (J). DNA is stained with DAPI. Scale bars, 20 μm.

Data information: All blots are representative of at least three independent experiments.
Source data are available online for this figure.

functionally active when bound to FAM83D in mitosis. To test this, we employed an IP-based kinase assay strategy to isolate the FAM83D–CK1α complex, and measure phosphorylation of an

optimised CK1 substrate peptide (CK1tide) using [γ-$^{32}$P]-ATP (Fig 5E). Excitingly, when anti-GFP IPs from asynchronous and mitotic extracts from *FAM83D*^−/−^, *FAM83D*^GFP/GFP^ and *FAM83D*^GFP/GFP(F283A)^

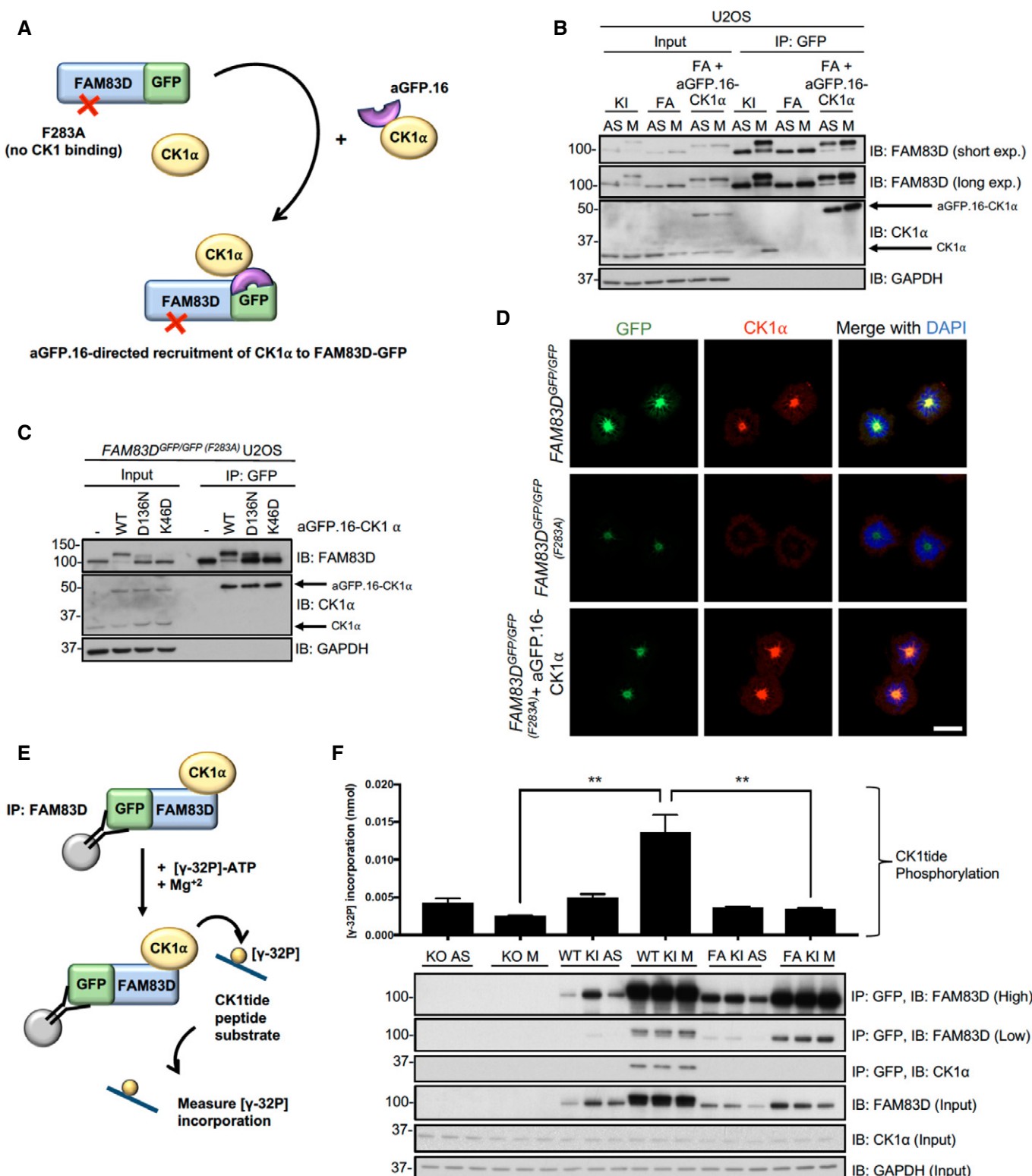

**Figure 5.**

cell lines were subjected to kinase assays against CK1tide, only those from mitotic *FAM83D*<sup>GFP/GFP</sup> extracts, which co-precipitated endogenous CK1α, displayed significant phosphorylation of CK1tide (Fig 5F).

**The FAM83D–CK1α complex regulates spindle positioning**

Separately, FAM83D and CK1α have been implicated in mitotic resolution and spindle/chromosome alignment [16,17,29,30]. So, we

**Figure 5. The role of CK1α in FAM83D phosphorylation.**

A  Schematic representation of the anti-GFP nanobody (aGFP.16)-based targeting strategy used to deliver CK1α to the CK1-binding-deficient FAM83D(F283A)-GFP mutant.
B  STLC-synchronised mitotic $FAM83D^{GFP/GFP}$ knockin (KI), $FAM83D^{GFP/GFP\ (F283A)}$ (FA) and $FAM83D^{GFP/GFP(F283A)}$ stably expressing aGFP.16-CK1α (FA + aGFP.16-CK1α) U2OS cells were subjected to GFP TRAP immunoprecipitation (IP), followed by immunoblotting (IB) with the indicated antibodies. Asynchronous (AS) cells were used as controls.
C  FA cells were infected with retroviruses encoding wild-type aGFP.16-CK1α (WT), or aGFP.16-CK1α with one of two distinct CK1α kinase-inactive mutants (D136N or K46D). Uninfected cells (−) were included as a control. Cells were lysed, subjected to anti-GFP IP and IB with the indicated antibodies.
D  Immunofluorescence analysis for the cells described in (B) following synchronisation with STLC. Cells were stained using anti-CK1α antibody, and DNA is stained with DAPI. Scale bars, 20 μm.
E  Schematic depicting the IP kinase assay strategy used to test whether FAM83D-bound CK1α was catalytically active.
F  $FAM83D^{-/-}$ knockout (KO), KI and FA cells were synchronised in mitosis (M) using STLC. Lysed extracts were subjected to anti-GFP IPs, followed by $[\gamma^{32}P]$-ATP kinase assays, using an optimised CK1 substrate peptide (CK1tide). AS cells were used as controls. Error bars, SEM; $P < 0.01$; ANOVA. $n = 3$. Input and IP samples were analysed by IB with the indicated antibodies.

Data information: All blots are representative of at least three independent experiments.
Source data are available online for this figure.

sought to test whether FAM83D and CK1α act in the same pathway in mitosis. Time lapse microscopy (Fig 6A) showed that, compared to wild-type and $FAM83D^{GFP/GFP}$ knockin cells, there was a significant metaphase-to-anaphase transitional delay in the $FAM83D^{-/-}$ and $FAM83D^{GFP/GFP(F283A)}$ cell lines (Fig 6B and C). When we analysed spindle orientation in these cells, we noted that in subconfluent $FAM83D^{-/-}$ and $FAM83D^{GFP/GFP(F283A)}$ cells, which are both deficient for spindle localisation of CK1α, spindles oriented at a fixed angle (Fig 6D, yellow lines), which is consistent with the phenotype described following siRNA depletion of $FAM83D$ [16]. The position of the cell division axis at anaphase, relative to its expected position aligned with the long cell axis in interphase, showed significant deviation in $FAM83D^{-/-}$ and $FAM83D^{GFP/GFP(F283A)}$ cells compared to wild-type and $FAM83D^{GFP/GFP}$ cells (Fig 6D and E). Excitingly, this phenotype in $FAM83D^{-/-}$ cells was rescued when $FAM83D$ cDNA was restored at the endogenous $FAM83D$ locus, with full rescue observed when restored FAM83D expression was comparable to that of wild-type cells (rescue clone 6) and a partial rescue observed when restored FAM83D expression was lower than that of wild-type cells (rescue clone 11) (Fig 6D and E). To better study the process of spindle orientation in individual cells, we seeded individual wild-type, $FAM83D^{-/-}$, $FAM83D^{GFP/GFP}$ or $FAM83D^{GFP/GFP(F283A)}$ U2OS cells onto L-shaped fibronectin-coated micropatterns, which cause cells to position their spindle on a defined axis [31], and measured the spindle orientation angles. Whereas wild-type and $FAM83D^{GFP/GFP}$ cells orientated their spindles along the defined, predicted axes, we observed significant deviations in the predicted spindle orientation axes with $FAM83D^{-/-}$ and $FAM83D^{GFP/GFP(F283A)}$ cells (Fig 6F and G), with nearly 80% unable to orientate their spindles correctly (Fig 6H) (for representative movies for each cell line, see Movies EV1–EV8). Again, this phenotype in the $FAM83D^{-/-}$ cells was fully rescued in $FAM83D$ cDNA knockin rescue clone 6, and partially in clone 11 (Fig 6G and H).

During the process of spindle positioning analyses, we observed a random orientation of the initial spindle position (Fig 6G), which is normally directed by the position of retraction fibres [32] and the assembly of subcortical actin [33,34]. To monitor actin dynamics during the process of spindle assembly and orientation, cells were transfected with actin-Red Fluorescent Protein (RFP) prior to seeding on L-shaped micropatterns. In wild-type cells, we observed polarised subcortical actin adjacent to the right angle of the L-shape following DNA condensation through to anaphase (Fig 6I).

However, actin appeared to be randomly organised with respect to the micropattern in $FAM83D^{-/-}$ and $FAM83D^{GFP/GFP(F283A)}$ cells during the spindle assembly process (Fig 6J). This observation in $FAM83D^{-/-}$ cells was rescued in $FAM83D$ cDNA knockin clone 6, and partially in clone 11 (Fig 6J). $FAM83D^{GFP/GFP}$ cells organised their actin in a manner comparable with wild-type cells (Fig 6J).

The targeted delivery of wild-type (WT) or kinase-dead (KD) CK1α to FAM83D(F283A)-GFP in knockin cells, notwithstanding the potential caveats of overexpression, allowed us to explore the role of CK1α catalytic activity in metaphase length and spindle orientation. Delivery of both WT and KD aGFP.16-CK1α resulted in shortening of the metaphase delay observed in $FAM83D^{GFP/GFP(F283A)}$ cells (Fig 6A–C). However, the spindle orientation defect in $FAM83D^{GFP/GFP(F283A)}$ cells was rescued completely with aGFP.16-CK1α, whereas only partially with the kinase-dead aGFP.16-CK1α, in both subconfluent cultures (Fig 6D and E) and L-shaped micropatterns (Fig 6G and H). Actin was correctly polarised in $FAM83D^{GFP/GFP(F283A)}$ cells rescued with WT, but not with the KD aGFP.16-CK1α (Fig 6J). Interestingly, we observed similar spindle positioning and actin cytoskeletal defects when CK1α was knocked down using siRNA oligonucleotides (Appendix Fig S1), reinforcing the notion that FAM83D and CK1α act together in mitosis.

We also observed frequent plasma membrane blebbing on one of the daughter cells in $FAM83D^{-/-}$ and $FAM83D^{GFP/GFP(F283A)}$ cells at the latter stages of mitosis (red arrows in Fig 6F). This phenomenon, known as asymmetric membrane elongation (AME), is a compensatory mechanism to ensure equal distribution of cell size following mitosis, when the spindle is misorientated [35]. Live cell microscopy confirmed blebbing in $FAM83D^{-/-}$ and $FAM83D^{GFP/GFP(F283A)}$ cells but not in wild-type, or $FAM83D^{GFP/GFP}$ cells, or in both clones rescued through knockin of $FAM83D$ cDNA into the endogenous $FAM83D$ locus in $FAM83D^{-/-}$ background (Appendix Fig S2). Reduction of blebbing in $FAM83D^{GFP/GFP(F283A)}$ cells to levels seen in wild-type cells was observed when cells were rescued with WT aGFP.16-CK1α, whereas an intermediate phenotype was observed with the KD aGFP.16-CK1α rescue (Appendix Fig S2). Importantly, there was no difference in daughter cell size between all cell lines (Appendix Fig S2). Taken together, these data indicate the FAM83D–CK1α interaction is critical for timely mitotic progression, including the processes of establishing and orienting both the mitotic spindle and the cell division axis.

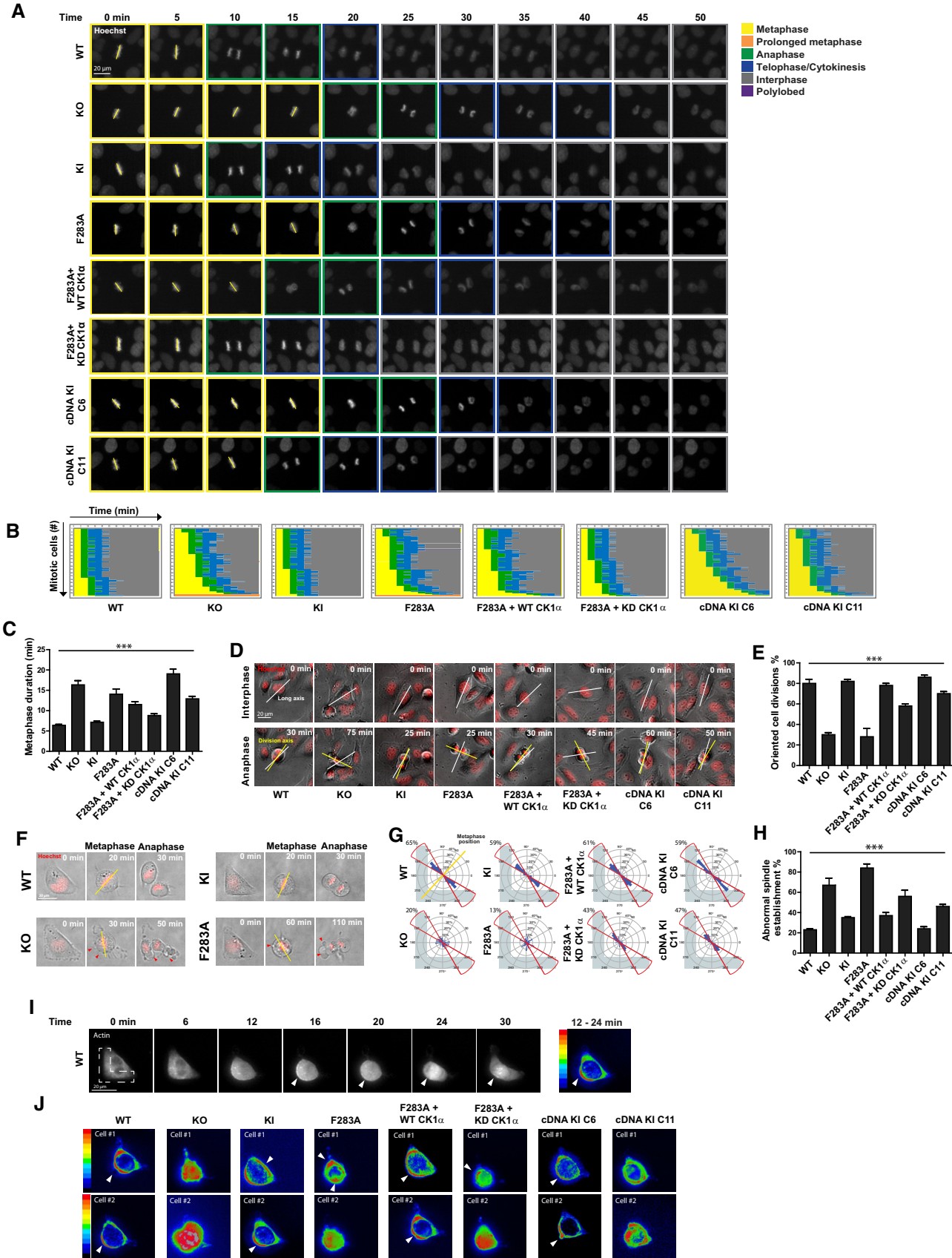

**Figure 6.**

**Figure 6. The FAM83D–CK1α complex regulates spindle positioning.**

A   Representative images initiating at metaphase for mitotic U2OS cell lines stained with Hoechst, taken every 5 min as they progress through division. Mitotic stage was determined by chromosome condensation and is indicated by the coloured boxes. Scale bar, 20 μm.

B   Graphical representation for the kinetics of transition from metaphase alignment (yellow) to anaphase (green), and cytokinesis (blue). 100 mitotic cells per genotype are plotted; *n* = 2.

C   Length of time needed to transition from metaphase to anaphase. Mean of 100 mitotic cells per genotype are plotted; *n* = 2. Error bars, SEM. ***$P < 0.0001$; ANOVA.

D   Representative bright-field images for mitotic U2OS cells indicating the long axis during interphase (white line) preceding mitosis, and the division axis determined at anaphase (yellow line). Scale bar, 20 μm.

E   Percentage of oriented divisions for U2OS cells grown in subconfluent cultures. An oriented division axis was defined as being $< 30°$ removed from the long axis of the interphase cell. Mean of 50 cells per genotype are plotted, *n* = 2. Error bars, SEM. ***$P < 0.0001$; ANOVA.

F   Representative images of mitotic U2OS cells stained with Hoechst and grown on L-shaped micropatterns previously coated with fibronectin. The position of metaphase chromosomes, which is plotted in panel (H), is indicated (yellow line) as is the presence of cortical blebbing (red arrows). Scale bar, 20 μm.

G   Circular graphs, superimposed on L-shaped micropattern, show the distribution of cell division angles measured at anaphase. Angles for 100 U2OS cells are plotted per genotype, *n* = 2. Metaphase position is indicated (yellow line), and the percentages of division angles $± 15°$ from the expected axis (red line) are indicated.

H   Percentage of metaphase U2OS cells that align chromosomes outside of the expected axis (angles $± 15°$). Mean of 100 cells per genotype are plotted, *n* = 2. Error bars, SEM. ***$P < 0.0001$; ANOVA.

I   Representative images of RFP-actin localisation in mitotic U2OS cells grown on fibronectin-coated, L-shaped micropatterns, which is superimposed. Arrowheads indicate polarised cortical actin. Heatmap shows the intensity of RFP-actin localisation (ImageJ z-projection standard deviation) as the cell progresses from prophase to metaphase (12–24 min). Scale bar, 20 μm.

J   Heatmap additive intensities of RFP-actin localisation in two representative mitotic U2OS cells for each genotype grown on fibronectin-coated, L-shaped micropatterns. Arrowheads indicate polarised cortical actin.

## Discussion

Correctly orientated cell division, achieved through proper spindle positioning, is crucial for both normal development and maintenance of healthy tissues [36–39]. By conclusively placing CK1α at the mitotic spindle to ensure proper spindle orientation and timely passage through mitosis, our study adds a new paradigm to the phospho-control of mammalian cell division. Interestingly, the mechanism by which the cell cycle-regulated protein FAM83D mediates the delivery of CK1α to the mitotic spindle draws parallels with how TPX2 recruits and regulates Aurora A during mitosis. While interactors of FAM83D, including HMMR and DYNLL1, offer insights into how FAM83D localises to the mitotic spindle to recruit CK1α, the full extent of CK1α substrates that potentially mediate proper spindle positioning to ensure error-free progression through mitosis remain to be defined. Furthermore, exactly how the spindle-localised FAM83D–CK1α complex can regulate the spatially distinct, yet not unrelated, actin cytoskeleton remains to be determined. However, phosphorylation of distinct substrates by CK1α may trigger their translocation from the spindle to the actin network directly or indirectly. Nonetheless, the findings that the FAM83D–CK1α complex acts at the mitotic spindle add to the evidence for intricate subcellular regulation of CK1 isoforms, which are implicated in many cellular processes from Wnt signalling to the regulation of circadian rhythms, by FAM83 proteins [13].

Given the participation of CK1 isoforms in such diverse biological processes, it is perhaps not surprising that some studies have explored and reported on roles for CK1α in the cell division cycle. Specifically, injection of CK1α antibodies into developing mouse embryos resulted in a significant delay in the progression to the first mitotic cleavage [29] and injection of CK1α morpholinos in mouse oocytes resulted in meiotic chromosomal alignment and congression defects [30]. However, these studies lacked any mechanistic understanding of how CK1α carried out these functions, as did the studies that reported somewhat similar phenotypes resulting from siRNA knockdown of *FAM83D* in cells [16,17]. Interestingly, CK1 activity in *Caenorhabditis elegans* embryos was shown to regulate spindle positioning through regulation of cortical force generation [40].

However, as FAM83D and the other FAM83 proteins are not conserved in invertebrates, there is likely to be an alternative way in which CK1 regulates spindle positioning in invertebrates. Here, our findings that FAM83D binds and recruits CK1α to the mitotic spindle for proper spindle positioning illuminate mechanistic insights into the role and regulation of CK1α in mitosis, and establish that CK1α catalytic activity at the mitotic spindle is required for smooth and efficient cell division. In support of our findings, it is also interesting that observations of CK1α at the mitotic spindle have been noted by immunostaining [41] and large-scale mitotic spindle proteomic [42] studies.

Beyond CK1α, the phosphorylation-mediated control of spindle positioning has been attributed to several other protein kinases. Of note, the activity of CDKs, PLKs and Aurora kinases have all been shown to critically regulate spindle positioning [15,43–50]. Both their recruitment to the mitotic spindle and activities are tightly regulated [44,46,51,52]. In the case of PLK1, the centrosomal PLK1 pool regulates spindle positioning through the PLK1-dependent spindle positioning pathway, and acts to strip force-generating Dynein complexes from the cortex when the spindle pole gets too close to the cortex [37]. PLK1 itself is regulated downstream of CDK1 and Aurora A, illustrating some of the crosstalk evident between these mitotic kinases [53]. Thus, the concerted action of mitotic kinases likely functions within a tightly controlled phospho-network to ensure the correct and efficient orientation of the mitotic spindle.

Recently, lenalidomide-induced degradation of CK1α was shown to be effective in the treatment of pre-leukaemic human myelodysplastic syndrome (MDS) [54]. Similarly, genetic ablation of CK1α has been shown to activate the tumour suppressor p53, suggesting CK1α could make an anti-cancer target [55,56]. Yet, other studies have demonstrated that CK1 inhibitors stabilise β-catenin and activate Wnt signalling, which promotes cell proliferation [11,57,58]. Our findings would suggest that the anti-proliferative effects of CK1α inhibitors might be occurring, in part, through the inhibition of the mitotic FAM83D-CK1α pool. Often thought of as undruggable kinases due to their participation in multiple, critical cellular processes, a method of targeting CK1 isoforms at specific locations under certain conditions is thus warranted, yet is very challenging.

However, building on data shown here, it may transpire that targeting the FAM83D–CK1α interaction may prove a viable therapeutic approach aimed at inhibition of proliferation.

# Materials and Methods

### Plasmids

Recombinant DNA procedures were performed using standard protocols as described previously [23]. Human wild-type *FAM83D*, *CSNK1A1*, *CSNK1E* or appropriate mutants were sub-cloned into pcDNA5/FRT/TO or pBABED P vectors (pBABED P denotes a Dundee-modified version of the pBABE Puro vector). *FAM83D* constructs harbour a Green Fluorescence Protein (GFP) tag at the N-terminus where indicated. All constructs are available to request from the MRC-PPU reagents webpage (http://mrcppureagents.d undee.ac.uk), and the unique identifier (DU) numbers indicated below provide direct links to sequence information and the cloning strategy used. The following constructs were generated: pcDNA5-FRT/TO GFP-FAM83D (DU29092), pcDNA5-FRT/TO GFP-FAM83D (F283A) (DU29109), pcDNA5-FRT/TO GFP-FAM83D (D249A) (DU29110), pBABED P aGFP.16-CK1α (DU29403), pBABED P aGFP.16-CK1α (K46D) (DU29555), pBABED P aGFP.16-CK1α (D136N) (DU28707), pBABED P aGFP.16-CK1ε (DU29613) and pBABED P aGFP.16-CK1ε (D128N) (DU29629). The AdPROM constructs used in this study have been described previously [22,23]. Constructs were sequence-verified by the DNA Sequencing Service, University of Dundee (http://www.dnaseq.co.uk). For amplification of plasmids, 1 μl of the plasmid was transformed into 10 μl of *Escherichia coli* DH5α competent bacteria (Invitrogen) on ice, incubated at 42°C for 45 s and then returned to ice for 2 min, before plating on LB-agar medium plate containing 100 μg/ml ampicillin. LB plates were inverted and incubated for 16 h at 37°C. Following incubation, a single colony was picked and used to inoculate 4 ml of LB medium containing 100 μg/ml ampicillin. Cultures were grown for 18 h at 37°C in a bacterial shaker (Infors HT). Plasmid DNA was purified using a Qiagen mini-prep kit as per the manufacturer's instructions. The isolated DNA yield was subsequently analysed and quantified using a Nanodrop 1,000 spectrophotometer (Thermo Scientific).

For CRISPR/Cas9 gene editing, the following guide RNAs (gRNA) and donor constructs were generated: *FAM83D* knockout: sense gRNA (DU52007), antisense gRNA (DU52023). *FAM83D* C-terminal GFP knockin: sense gRNA (DU54048), antisense gRNA (DU54054), GFP donor (DU54198). *FAM83D* C-terminal GFP knockin with F283A mutation: sense gRNA (DU57831), antisense gRNA (DU57835), GFP donor (DU57512). *FAM83D* restoration in the knockout background: sense gRNA (DU60528), antisense gRNA (DU60530), *FAM83DcDNA-IRES-GFP-polyA* donor (DU60707). *CSNK1A1* N-terminal mCherry knockin: sense gRNA (DU57522), antisense gRNA (DU57527), mCherry donor (DU57578). *CSNK1E* N-terminal mCherry knockin: sense gRNA (DU54377), antisense gRNA (DU54383), mCherry donor (DU57623).

### Cell culture

Human osteosarcoma U2OS, cervical cancer HeLa, lung adenocarcinoma A549, keratinocyte HaCaT or mouse embryonic fibroblasts (MEF) cells were grown in Dulbecco's modified Eagle's medium (DMEM; Gibco) containing 10% (v/v) Foetal Bovine Serum (FBS; Hyclone), penicillin (100 U/ml; Lonza), streptomycin (0.1 mg/ml; Lonza) and L-glutamine (2 mM; Lonza), and cultured at 37°C, 5% $CO_2$ in a humidified tissue culture incubator. The wild-type and *HMMR* knockout MEFs were generated by the Maxwell laboratory and have been described previously [15]. MEFs were immortalised using simian virus 40 (SV40) retrovirus, to enable progression through the cell cycle. All cells used in this study were verified to be free from mycoplasma contamination. Cells were exposed to different stimuli and compounds as described in the appropriate figure legends prior to lysis. For transient transfections, cells were transfected for 24 h with 2 μg cDNA (per 10-cm dish) in serum free OptiMem (Gibco) with the transfection reagent polyethylenimine (PEI) as described previously [22,23,59]. For retroviral-based infections, cells were infected with retroviruses as described previously [22,23].

### Cell synchronisation

For synchronisation, cells were arrested at prometaphase with nocodazole (100 ng/ml) for 12 h, before floating mitotic cells were isolated through mitotic shake-off. Collected mitotic cells were washed 3× in PBS before re-plating in fresh full medium for 45 min before lysis, to allow them to progress into mitosis. Alternatively, cells were synchronised in mitosis using the Eg5 inhibitor S-trityl L-cysteine (STLC) [18] (5 μM) for 16 h. Following incubation, mitotic cells were isolated through shake-off and were either washed 3× in PBS and re-plated into fresh medium, or washed 2× in ice-cold PBS, and lysed. Where appropriate, MG132 and ProTAME were used at 20 μM final concentration. For G2 arrest, cells were treated with 10 μM of the CDK1 inhibitor RO-3306 [21] for 24 h before lysis. To trigger forced mitotic exit once cells were already in mitosis, cells were treated with 10 μM RO-3306 for 1 h prior to lysis.

### Live cell imaging

U2OS cells were grown in 96-well plates (Corning) and imaged for up to 24 h at 37°C in a 5% $CO_2$ environmental chamber using a 40× 0.75 NA dry objective with the MetaXpress 5.0.2.0 software (Molecular Devices Inc.) on the ImageXpress Micro XL epifluorescence microscope (Molecular Devices Inc.). For the analysis of cell division kinetics, cells were stained with Hoechst (1 μg/ml) to label DNA and images were taken every 5 min, and movies were made in the MetaXpress 5.0.2.0 software (Molecular Devices Inc.). For actin localisation, U2OS cells were seeded at 20% confluency in 24-well plates, and 1 μl CellLight Actin-RFP (Thermo Fisher Scientific) was added per 5,000 cells and incubated at 37°C for 16 h. Following the incubation, U2OS cells were seeded on L-shaped micropatterns (CYTOO) at 30,000 per ml. For the analysis of actin localisation, images were taken every 2 min, and movies were made in the MetaXpress 5.0.2.0 software (Molecular Devices Inc.). Images of actin localisation were projected from prophase to metaphase (ImageJ, z-projection standard deviation) for analysis. For the analysis of cortical membrane elongation, images were taken every 1 min, and movies were made in the MetaXpress 5.0.2.0 software (Molecular Devices Inc.).

## Quantification of spindle orientation

U2OS cells were seeded at 3,000 cells per well in 96-well plates with L-shaped micropatterns (CYTOO) at a density of 15,000 cells/ml. Prior to seeding, plates were coated with 20 μg/ml fibronectin (Sigma) for 2 h at RT. Following seeding, cells were imaged every 5 or 10 min for up to 24 h at 37°C in a 5% $CO_2$ environmental chamber using an ImageXpress Micro High Content Screening System (Molecular Devices Inc.). To measure spindle orientation in subconfluent cultures, cells were seeded at 50% confluency, grown overnight and imaged as previously described [60]. Spindle angles were measured using a vector drawn through the division axis at anaphase bisecting a vector drawn through the cell's long axis determined prior to prophase.

## Flow cytometry

For cell cycle distribution profiles, U2OS cells treated with or without synchronisation agents as described above were collected and washed 2× in PBS + 1% (v/v) FBS. Cells were fixed in 90% (v/v) ice-cold methanol for either 20 min or O/N at −20°C. Following fixation, cells were washed 2× in PBS + 1% FBS and stained with DNA staining buffer (50 μg/ml propidium iodide, 50 μg/ml RNase A, in PBS + 1% FBS). Following 20-min incubation at RT protected from light, samples were analysed and data acquired on a fluorescence-activated cell sorting (FACS) Canto [Becton Dickinson, (BD)] using BD FACSDIVA Software. Data were visualised using FlowJo software (Tree Star, BD). Pulse-width analysis was used to ensure the exclusion of doublets and clumps prior to evaluation of cell cycle distribution using the Watson-Pragmatic model.

## Generation of $FAM83D^{-/-}$ knockout, $FAM83D^{GFP/GFP}$, $FAM83D^{GFP/GFP(F283A)}$, FAM83DcDNA-rescue cells, $^{mCherry/mCherry}CSNK1A1$ and $^{mCherry/mCherry}CSNK1E$ knockin cells using CRISPR/Cas9

To generate $FAM83D^{-/-}$ knockout by CRISPR/Cas9 genome editing, U2OS cells were transfected with vectors encoding a pair of guide RNAs (pBABED-Puro-sgRNA1 and pX335-CAS9-D10A-sgRNA2) targeting around the first exon of *FAM83D* (1 μg each). For GFP knockins, U2OS cells were transfected with vectors encoding a pair of guide RNAs (pBABED-Puro-sgRNA1 and pX335-CAS9-D10A-sgRNA2) targeting around the stop codon of *FAM83D* ($FAM83D^{GFP/GFP}$) or the region surrounding the codon encoding amino acid F283 ($FAM83D^{GFP/GFP(F283A)}$), along with the respective donor plasmid carrying the *GFP* knockin insert and flanking homology arms (~500 bases) (3 μg each). For the $FAM83D^{GFP/GFP(F283A)}$ knockin, the 5′ homology arm of the *GFP* donor was extended to 1,000 bp, in order to cover the desired site of mutation. For mCherry knockins, $FAM83D^{GFP/GFP}$ cells were transfected with vectors encoding a pair of guide RNAs (pBABED-Puro-sgRNA1 and pX335-CAS9-D10A-sgRNA2, 1 μg each) targeting around the start codon of *CSNK1A1 or CSNK1E*, along with the respective donor plasmid carrying the *mCherry* knockin insert and flanking homology arms (~500 bases) (3 μg each). To restore *FAM83D* in the knockout environment, we transfected the $FAM83D^{-/-}$ cells with a pair of guide RNAs (pBABED-Puro-sgRNA1 and pX335-CAS9-D10A-sgRNA2) targeting around the start codon of *FAM83D*, which had previously been

targeted for generation of $FAM83D^{-/-}$ knockout cells, along with the respective donor plasmid carrying the *FAM83DcDNA-IRES-GFP-polyA* knockin insert with flanking homology arms (~500 bases) (3 μg each). 16 h post-transfection, cells were selected in puromycin (2 μg/ml) for 48 h. The transfection process was repeated one more time. For the acquisition of single-cell clones of knockouts, and GFP and mCherry knockins, single cells were isolated by fluorescence-activated cell sorting (FACS) using an Influx cell sorter (Becton Dickinson). Single cell clones were plated on individual wells of two 96-well plates, pre-coated with 1% (w/v) gelatin to help cell adherence. Viable clones were expanded, and successful knockout or integration of *GFP*, *mCherry* or *FAM83D* cDNA at the target locus was confirmed by both Western blotting and genomic DNA sequencing.

## Generation of control and VHL-aGFP.16 AdPROM cell lines

Control and VHL-aGFP.16 AdPROM cell lines were generated as described previously [22,23], using retroviral-based infections to deliver the AdPROM constructs to the $FAM83D^{GFP/GFP}$ U2OS cells.

## RNA interference

U2OS cells were transfected with either ON-TARGETplus Non-targeting Control siRNAs (D-001810-01-05, Dharmacon) or ON-TARGETplus SMARTpool-Human siRNA targeting *CSNK1A1* (l-003957-00-0005, Dharmacon) using Lipofectamine 3000 Reagent (L3000015, Thermo Fisher). Lipid–siRNA mixtures were incubated with cells for 16 h, and then, cells were washed. Cells were collected for analysis after a further 48-h incubation post-transfection.

## Cell lysis and immunoprecipitation

Cells were washed twice in ice-cold PBS, before scraping/harvesting on ice in lysis buffer (50 mM Tris–HCl pH 7.5, 0.27 M sucrose, 150 mM NaCl, 1 mM EGTA, 1 mM EDTA, 1 mM sodium orthovanadate, 10 mM sodium β-glycerophosphate, 50 mM sodium fluoride, 5 mM sodium pyrophosphate and 1% Nonidet P40 substitute), supplemented with 1× cOmplete™ protease inhibitor cocktail (Roche). Cell extracts were either clarified and processed immediately, or snap frozen in liquid nitrogen and stored at −80°C. Protein concentrations were determined in a 96-well format using the Bradford protein assay reagent (Bio-Rad).

For Immunoprecipitations (IPs), clarified extracts were normalised in lysis buffer to typically 1–5 mg/ml. After input aliquots were collected, lysates were incubated for 4 h or O/N at 4°C with protein G-sepharose beads coupled to the antibody of interest, on a rotating wheel. For anti-FLAG IPs, FLAG M2 resin (Sigma) was used; for anti-GFP IPs, GFP TRAP beads (ChromoTek) were used; and for anti-mCherry IPs, RFP TRAP beads (ChromoTek) were used. For anti-FAM83D IPs, anti-FAM83D-coupled sepharose beads were used, and for anti-CK1α IPs, anti-CK1α-coupled sepharose beads were used, using our in house generated anti-FAM83D and anti-CK1α antibodies, respectively. Sheep IgG-coupled sepharose beads were employed as a control for endogenous IPs. Following incubation, beads were pelleted and flow-through extracts collected. Beads were washed once in lysis buffer supplemented with 250 mM NaCl, and 2–3 times in lysis buffer. For elution, beads were resuspended in 1× SDS sample buffer and incubated at 95°C for 5 min.

For mass spectrometry, IPs were performed as described above except that, prior to incubation with the relevant antibody-coupled beads, extracts were pre-cleared by incubating with protein G-sepharose beads for 1 h at 4°C on a rotating wheel. For elution, samples were boiled at 95°C for 5 min in 1× SDS sample buffer and eluted by spinning through SpinX columns (Corning).

## SDS–PAGE and Western blotting

Reduced protein extracts (typically 10–20 μg protein) or IPs were resolved on either 8 or 13% (v/v) SDS–PAGE gels, or 4–12% NuPAGE bis-tris precast gradient gels (Invitrogen) by electrophoresis. Separated proteins were subsequently transferred onto polyvinylidene fluoride (PVDF) membranes (Millipore), before membranes were blocked in 5% (w/v) non-fat milk powder (Marvel) in TBS-T (50 mM Tris–HCl pH 7.5, 150 mM NaCl, 0.2% (v/v) Tween-20) and incubated overnight at 4°C in either 5% milk TBS-T or 5% bovine serum albumin (BSA) TBS-T with the appropriate primary antibody. Membranes were then washed for 3 × 10 min with TBS-T before incubating with HRP-conjugated secondary antibodies in 5% milk TBS-T for 1 h at RT. Membranes were then washed for 3 × 10 min with TBS-T before detection with enhanced chemiluminescence reagent (Millipore) and exposure to medical-grade X-ray films (Konica Minolta), as described previously [20,59,61]. Alternatively, membranes were imaged using the Chemi-Doc™ system (Bio-Rad).

## Antibodies

The antibodies used in this study are listed in Table 1. The source, catalogue number and dilution factor used for each antibody are provided. Additionally, we provide references for the antibody specificity tests and highlight which antibody was used in each figure. For non-commercially sourced antibodies, antibodies were generated by the Division of Signal Transduction Therapy (DSTT), University of Dundee as described previously [20,59]. For HRP-coupled secondary antibodies, goat anti-rabbit-IgG (cat.: 7074, 1:2,500) was from CST, and rabbit anti-sheep-IgG (cat.: 31480, 1:5,000), goat anti-rat IgG (cat.: 62-9520, 1:5,000) and goat anti-mouse-IgG (cat.: 31430, 1:5,000) were from Thermo Fisher Scientific.

For immunofluorescence studies, signal amplification was achieved using AlexaFluor-594-conjugated donkey anti-sheep IgG (H+L) (cat.: A11058, Life Technologies, 1:300), AlexaFluor-594-conjugated goat anti-rabbit IgG (H+L) (cat.: A11012, Life Technologies, 1:500) and goat anti-mouse IgG (H+L) (cat.: A11005, Life Technologies, 1:500).

## Fluorescence microscopy

Cells were seeded onto glass coverslips and treated/transfected as described above or in figure legends. Cells were washed 2× in PBS, before fixing in 4% (w/v) paraformaldehyde (PFA) for 20 min at RT. Cells were washed 2× in DMEM/10 mM HEPES, followed by incubation in DMEM/10 mM HEPES for 10 min. Cells were washed once in PBS and permeabilised for 3 min in 0.2% NP-40. Following permeabilisation, cells were washed 2× in PBS containing 1% (w/v) BSA, followed by incubation in PBS/BSA for 15 min. Where appropriate, coverslips were then incubated with primary antibody in PBS/BSA (typically at 1:100-1:500 dilution) at 37°C for 1–1.5 h. Cells were washed for a minimum of 3 × 10 min in PBS/BSA before incubation with the secondary AlexaFluor-conjugated antibody in PBS/BSA (1:300–500 dilution) for 60 min at RT protected from light. Coverslips were subsequently washed for 3 × 10 min in PBS/BSA and mounted on glass microscopy slides using ProLong® Gold anti-fade reagent with DAPI (Life Technologies). Coverslips were sealed with clear nail varnish and left to dry overnight before analysis on a Zeiss LSM710 confocal microscope using a 63× Plan-Apochromat objective (NA 1.40). Alternatively, cells were imaged on a Nikon TiE inverted microscope (60× objective) and visualised with NIS Elements (Nikon). Images were processed with Omero [62].

For quantification of CK1α spindle localisation, the mean pixel intensities of CK1α staining on the spindle were calculated by measuring the mean pixel intensities of CK1α in the region of interest (roi) demarcated by the outer border of the DAPI ring (hereafter referred to as the spindle roi). Subsequently, the ratio between the spindle roi and the background cytoplasmic CK1α staining was calculated, by measuring the mean cytoplasmic CK1α pixel intensities in the cytoplasmic roi, defined as the whole cell minus the spindle roi. The resulting ratios were plotted on a box plot with whiskers indicating the highest and lowest values. A ratio of > 1 indicates CK1α is present on the spindle, and a ratio of ≤ 1 indicates CK1α is not present on the spindle. One might expect the theoretical ratio to be close to 1 in the $FAM83D^{-/-}$ cells. However, lower mean CK1α intensity in the DAPI-stained area relative to the cytoplasm implies exclusion of CK1α in DNA-rich regions. This results in a lower mean CK1α staining intensity in the overall spindle roi versus the cytoplasmic roi and therefore a ratio of < 1. The ImageJ macro developed by Graeme Ball (Dundee Imaging Facility), used for this purpose, is included as a supplementary file (Appendix File S1).

## Mass spectrometry

Proteins were affinity purified from clarified extracts by GFP TRAP beads (ChromoTek) as described above. Purified proteins were resolved on 4–12% gradient gels by SDS–PAGE, and gels were subsequently stained with InstantBlue™ (Expedeon). Gel slices covering each lane were excised and digested with trypsin. The tryptic peptides were subjected to mass spectrometric analysis performed by LC-MS-MS on a Linear ion trap-orbitrap hybrid mass spectrometer (Orbitrap-VelosPro, Thermo) coupled to a U3000 RSLC HPLC (Thermo). Peptides were trapped on a nanoViper Trap column, 2 cm × 100 μm C18 5 μm 100 Å (Thermo, 164564), then separated on a 15 cm Thermo EasySpray column (ES800) equilibrated with a flow of 300 nl/min of 3% Solvent B [Solvent A: 2% Acetonitrile, 0.1% formic acid, 3% DMSO in $H_2O$; Solvent B: 80% acetonitrile, 0.08% formic acid, 3% DMSO in $H_2O$]. The elution gradient was as follows: Time(min):Solvent B(%); 0:3, 5:3, 45:35, 47:99, 52:99, 55:3, 60:3. Data were acquired in the data-dependent mode, automatically switching between MS and MS-MS acquisition. Full scan spectra ($m/z$ 400–1,600) were acquired in the orbitrap with resolution R = 60,000 at $m/z$ 400 (after accumulation to an FTMS Full AGC Target; 1,000,000; FTMS MSn AGC Target; 50,000). The 20 most intense ions, above a specified minimum signal threshold (2,000), based upon a low resolution (R = 15,000) preview of the survey scan, were fragmented by collision induced dissociation

**Table 1.  Antibodies used in this study.**

| Antigen | Source | Cat. Number | Dilution | Used in Figure | Specificity Ref. |
|---------|--------|-------------|----------|----------------|------------------|
| CK1α | Bethyl | A301-991A | WB: 1:1,000 | 1D E, G, H and I, and 2D and 4B–D and 5F and EV3A and B, Appendix Fig S1A | 12 |
| CK1α | DSTT (University of Dundee) | Sheep SA527, third bleed | WB: 1:1,000; IF: 1:1,000 | 3B and F, and 4G and I, and 5B and C, and EV4A–C | This study |
| CK1α | Santa Cruz | sc-6477 | IF: 1:100 | 2A, B and G, and 3D and 5D and EV3C | 12 |
| CK1ε | CST | 12448 | WB: 1:1,000 | 1D and EV5B | 12 |
| CK1ε | DSTT (University of Dundee) | Sheep SA610, second bleed | WB: 1:1,000 | EV4A–C | This study |
| CK1ε | Sigma-Aldrich | HPA026288 | IF: 1:500 | EV2A | 12 |
| CK1δ | CST | 12417 | WB: 1:1,000 | 1D | 12 |
| Cyclin A2 | CST | 4656 | WB: 1:1,000 | 1I | This study |
| Cyclin B1 | CST | 4138 | WB: 1:1,000 | 1D, E and I, and 4A–D | This study |
| DYNLL1 | Abcam | EP1660Y | WB: 1:1,000 | 1D | This study |
| FAM83D (C-term) | DSTT (University of Dundee) | Sheep SA102, third bleed | WB: 1:1,000; IF: 1:500 | 1A, D, E, H and I, and 2D and F, and 3B, C and F and 4A–D and 5B, C and F, and EV3A and B, and EV4B and EV5B | This study |
| FAM83D (N-term) | DSTT (University of Dundee) | Sheep SA102, third bleed | WB: 1:1,000 | 1A and G, and 4G | This study |
| FAM83G | DSTT (University of Dundee) | Sheep S876C, third bleed | WB: 1:1,000 | EV4C | 12,14 |
| GAPDH | CST | 14C10 | WB: 1:5,000 | 1A, D, E, G, H and I, and 2D and F, and 4A–D and G, and 5B, C and F, and EV3A and B, and EV4A | |
| GAPDH | Proteintech | 10494-01-AP | WB: 1:5,000 | Appendix Fig S1A | |
| GFP | Roche | 11814460001 | WB: 1:500 | 3B | This study |
| HMMR | Millipore | ABC323 | WB: 1:1,000; IF: 1:500 | 1D, E and I, and 2D and 4B–D and EV2B and EV3A and B | This study |
| HMMR | Abcam | 124729 | WB: 1:1,000; IF: 1:500 | 4G and H | This study |
| p-Histone H3 (S10) | CST | 9701 | WB: 1:1,000 | 4B | This study |
| VHL | CST | 68547 | WB: 1:1,000 | 2F | 22,23 |

CST, cell signaling technology; DSTT, division of signal transduction therapy; WB, Western blotting; IF, immunofluorescence.

All the antibodies used in this study are listed, along with their source, catalogue number and dilution factor used. References for antibody specificity and details of the figures in which each antibody was used are also provided.

and recorded in the linear ion trap (Full AGC Target; 30,000. MSn AGC Target; 5,000). Data files were analysed by Proteome Discoverer 2.0 (www.ThermoScientific.com), using Mascot 2.4.1 (www.matrixscience.com), and searching the SwissProt Human database. Scaffold Q/Q+S V4.4.7 (www.ProteomeSoftware.com) was also used to examine the Mascot result files. Allowance was made for the following fixed, carbamidomethyl (C) and variable modifications, oxidation (M), and dioxidation (M). Error tolerances were 10 ppm for MS1 and 0.6 Da for MS2. Scaffold Q/Q+S V4.4.6 (www.ProteomeSoftware.com) was used to further analyse the data and obtain values for the total unique peptide counts for each protein.

For the qualitative analysis of FAM83D-interacting proteins, we employed a strict set of requirements when determining whether a protein was likely to interact with FAM83D in asynchronous (AS), mitotic (M) or both AS and M conditions. For both nocodazole and STLC treatments, only proteins, which were identified by at least five total unique peptides, were included in the analysis. Crucially, there must have been > 5 total unique peptides between the negative control and AS or M samples for a protein to be considered as a non-contaminant. To be deemed as an AS- or M-specific FAM83D-interacting protein, there must have been greater than 10 total unique peptides between the AS and M samples.

### RNA-isolation, cDNA synthesis and qRT–PCR

Cells were washed in PBS, and RNA was isolated following the manufacture's guidelines (Qiagen). cDNA synthesis and qRT–PCR were performed as described previously [14,22]. The following primer pairs were used: FAM83D (Forward: ACGTTGATTGATGG CATCCG; Reverse: CCTTGGACTGTGGTTTTCGG), HMMR (Forward: CAAAAGAGAAACAAAGATGAG-GGG; Reverse: CCACTTGATCTGA AGCACAAC), CK1α (Forward: AATGTTAAAG-CAGAAAGCAGCAC; Reverse: TCCTCAATTCATGCTTAGAAACC), cyclin B1 (Forward:

GCAGTGC-GGGGTTTAAATCT; Reverse: GCCATGTTGATCTTCGCCTT) and GAPDH (Forward: TGCACCACCAACTGCTTAGC; Reverse: GGC ATGGACTGTGGTCATGAG). Primers were obtained from Invitrogen.

### Phosphatase assays

Lysed extracts were subjected to GFP TRAP immunoprecipitation and washed 3× in wash buffer (200 mM NaCl, 50 mM Tris–HCl pH 7.5, 1% Triton X-100). Beads were resuspended in 20 μl phosphatase-assay buffer [New England Biolabs (cat.: PO753)] containing 1 mM $MnCl_2$, with or without λ-phosphatase (1 U) for 30 min at 30°C, with shaking. Following incubation, beads were washed 3× in wash buffer and eluted. Input and immunoprecipitation samples were subjected to Western blotting as described above.

### Purification of recombinant proteins

Most recombinant proteins used in the *in vitro* kinase assays were purified by the Division of Signal Transduction Therapy (DSTT; University of Dundee), and the identities of the expressed proteins verified by mass spectrometry. Each protein has a unique identification number to request from the MRC-PPU Reagents website (http://mrcppureagents.dundee.ac.uk) as follows: GST-CK1α (DU329) and GST-FAM83D (DU28270). GST-FAM83G-6xHis was purified by Polyxeni Bozatzi [14]. Briefly, the proteins were expressed in BL21 (DE3) *E. coli* as described previously [12], and affinity purified using GSH-sepharose or Nickel-agarose columns as appropriate.

### Kinase assays

For peptide-based kinase assays, reactions were set up and performed as described by Hastie *et al* [63] except that FAM83D-GFP IPs, in which endogenous CK1α co-immunoprecipitated, or mCherry-CK1α/ε IPs, were used instead of recombinant proteins. An optimised CK1 peptide [CK1tide (KRRRALS*VASLPGL), where S* indicates phospho-Ser] was used as the substrate. Assays were performed using samples from three biological replicates.

For recombinant substrate-based kinase assays, 25-μl reactions containing 200 ng of kinase (GST-CK1α) and 2 μg of substrate (Precision protease-cleaved FAM83D, initially expressed as GST-FAM83D) in a buffer composed of 50 mM Tris–HCl (pH 7.5), 0.1 mM EGTA, 10 mM magnesium acetate, 2 mM DTT and 0.1 mM $[\gamma^{32}P]$-ATP (500 cpm/pmol). Following 30-min incubation at 30°C, assays were stopped by adding 9 μl of 4× SDS sample buffer containing 5% (v/v) β-mercaptoethanol, with subsequent heating at 95°C for 5 min. Samples were resolved by SDS–PAGE, and the gels were stained with InstantBlue (Expedeon) and dried. Radioactivity was analysed by autoradiography.

### Statistical analysis

For qRT–PCR data, GraphPad (Prism) was used to generate plots and analyse data by unpaired Student's *t*-test. A *P*-value of < 0.05 was deemed significant. For quantification of CK1α spindle localisation, kinase assays and spindle orientation studies, GraphPad (Prism) was used to generate plots and analyse data by one-way ANOVA, followed by a post hoc Tukey's test. A *P*-value of < 0.05 was deemed significant.

## Data availability

The mass spectrometry proteomics data have been deposited to the ProteomeXchange Consortium via the PRIDE partner repository with the dataset identifier PXD013808 (https://www.ebi.ac.uk/pride/archive/projects/PXD013808).

**Expanded View** for this article is available online.

## Acknowledgements

We thank GS laboratory members, and A. Rojas-Fernandez, M. Muqit, J. Zomerdijk, T. Ly, J. Taylor, S. Virdee, A. Saurin and A. Musacchio for their highly appreciated experimental advice and/or discussions during the course of these experiments. We thank R. Filipe Soares for help with the PRIDE repository submission. We thank L. Fin, J. Stark and A. Muir for help and assistance with tissue culture, the staff at the DNA Sequencing services (School of Life Sciences, University of Dundee), and the cloning, antibody and protein production teams within the MRC-PPU reagents and services (University of Dundee), coordinated by J. Hastie and H. McLauchlan. We thank the staff at the Dundee Imaging Facility (School of Life Sciences, University of Dundee), and the staff at the flow cytometry facility (School of Life Sciences, University of Dundee) for their invaluable help and advice throughout this project. LJF is supported by the U.K. MRC PhD studentship. The Dundee Imaging Facility is funded by the "MRC Next Generation Optical Microscopy" award [MR/K015869/1]. LJF also receives funding from the Queens College Scholarship, University of Dundee. CAM is supported by the Michael Cuccione Foundation and the Canadian Institutes of Health Research (New Investigator Salary Award and Operating Grant OBC_134038). GPS is supported by the U.K. MRC (Grant MC_UU_12016/3) and the pharmaceutical companies supporting the Division of Signal Transduction Therapy (Boehringer-Ingelheim, GlaxoSmithKline, Merck-Serono).

## Author contributions

LJF performed most experiments, collected and analysed data, and contributed to the writing of the manuscript. ZH and LM performed experiments for spindle orientation, mitosis and blebbing assays and analysed data. TJM designed strategies and developed methods for all of the CRISPR/Cas9 gene editing, in addition to generating most of the constructs, used in the study. TJM and NTW cloned genes and performed mutagenesis experiments. ARP performed imaging experiments. AJW and RC performed flow cytometry, cell sorting and subsequent analysis for the DNA distribution profiles. JV, RG and DGC performed mass spectrometry experiments, collected and analysed data. GB developed the ImageJ Macro for quantifying CK1α spindle localisation. CAM coordinated the spindle orientation, mitosis and blebbing assays, analysed data and contributed to the writing of the manuscript. GPS conceived the project, analysed data and contributed to the writing of the manuscript.

## Conflict of interest

The authors declare that they have no conflict of interest.

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
