## [Review Process File · EMBO Reports]

FAM83D directs protein kinase CK1 α to the mitotic spindle for proper spindle positioning

Luke J. Fulcher, Zhengcheng He, Lin Mei, Thomas J. Macartney, Nicola T. Wood, Alan R. Prescott, Arlene J. Whigham, Joby Varghese, Robert Gourlay, Graeme Ball, Rosemary Clarke, David G. Campbell, Christopher A. Maxwell, and Gopal P. Sapkota

Review timeline:	Submission date:	29 November 2018
	Editorial Decision:	16 January 2019
	Revision received:	29 April 2019
	Editorial Decision:	11 June 2019
	Revision received:	13 June 2019
	Accepted:	26 June 2019

Editor: Deniz Senyilmaz-Tiebe

Transaction Report:

1st Editorial Decision

16 January 2019

Thank you for submitting your manuscript for consideration by EMBO Reports. It has now been seen by two referees whose comments are shown below.

As you can see, both referees express interest in the proposed role of FAM83D and CK1 α in mitotic spindle positioning. However, they also raise concerns that need to be addressed in full before we can consider publication of the manuscript here.

Given these constructive comments, we would like to invite you to revise your manuscript with the understanding that the referee must be fully addressed and their suggestions taken on board. Please address all referee concerns in a complete point-by-point response. Acceptance of the manuscript will depend on a positive outcome of a second round of review. It is EMBO Reports policy to allow a single round of revision only and acceptance or rejection of the manuscript will therefore depend on the completeness of your responses included in the next, final version of the manuscript.

You can submit the revision either as a Scientific Report or as a Research Article. For Scientific Reports, the revised manuscript can contain up to 5 main figures and 5 Expanded View figures. If the revision leads to a manuscript with more than 5 main figures it will be published as a Research Article. If a Scientific Report is submitted, these sections have to be combined. This will help to

shorten the manuscript text by eliminating some redundancy that is inevitable when discussing the same experiments twice. In either case, all materials and methods should be included in the main manuscript file.

Supplementary/additional data: The Expanded View format, which will be displayed in the main HTML of the paper in a collapsible format, has replaced the Supplementary information. You can submit up to 5 images as Expanded View. Please follow the nomenclature Figure EV1, Figure EV2 etc. The figure legend for these should be included in the main manuscript document file in a section called Expanded View Figure Legends after the main Figure Legends section. Additional Supplementary material should be supplied as a single pdf labeled Appendix. The Appendix includes a table of content on the first page with page numbers, all figures and their legends. Please follow the nomenclature Appendix Figure Sx throughout the text and also label the figures according to this nomenclature. For more details please refer to our guide to authors.

When preparing your letter of response to the referees' comments, please bear in mind that this will form part of the Review Process File, and will therefore be available online to the community. For more details on our Transparent Editorial Process, please visit our website: http://emboj.embopress.org/about#Transparent_Process

Regarding data quantification, please ensure to specify the name of the statistical test used to generate error bars and P values, the number (n) of independent experiments underlying each data point (not replicate measures of one sample), and the test used to calculate p-values in each figure legend. Discussion of statistical methodology can be reported in the materials and methods section, but figure legends should contain a basic description of n, P and the test applied. Please also include scale bars in all microscopy images.

We now strongly encourage the publication of original source data with the aim of making primary data more accessible and transparent to the reader. The source data will be published in a separate source data file online along with the accepted manuscript and will be linked to the relevant figure. If you would like to use this opportunity, please submit the source data (for example scans of entire gels or blots, data points of graphs in an excel sheet, additional images, etc.) of your key experiments together with the revised manuscript. Please include size markers for scans of entire gels, label the scans with figure and panel number, and send one PDF file per figure.

- a complete author checklist, which you can download from our author guidelines (<http://emboj.embopress.org/authorguide#revision>). Please insert page numbers in the checklist to indicate where the requested information can be found.
 - a letter detailing your responses to the referee comments in Word format (.doc)
 - a Microsoft Word file (.doc) of the revised manuscript text
 - editable TIFF or EPS-formatted figure files in high resolution
- (In order to avoid delays later in the publication process please check our figure guidelines before preparing the figures for your manuscript:
http://www.embopress.org/sites/default/files/EMBOPress_Figure_Guidelines_061115.pdf)
- a separate PDF file of any Supplementary information (in its final format)
 - all corresponding authors are required to provide an ORCID ID for their name. Please find instructions on how to link your ORCID ID to your account in our manuscript tracking system in our Author guidelines (<http://emboj.embopress.org/authorguide>).

As part of the EMBO publication's Transparent Editorial Process, EMBO reports publishes online a Review Process File to accompany accepted manuscripts. This File will be published in conjunction with your paper and will include the referee reports, your point-by-point response and all pertinent correspondence relating to the manuscript.

You are able to opt out of this by letting the editorial office know (emboreports@embo.org). If you do opt out, the Review Process File link will point to the following statement: "No Review Process

File is available with this article, as the authors have chosen not to make the review process public in this case."

I look forward to seeing a revised version of your manuscript when it is ready. Please let me know if you have questions or comments regarding the revision.

REFEREE REPORTS

Referee #1:

Fulcher and coworkers identify a mitosis-specific interaction between FAM83D and CK1 α that recruits CK1 α to the mitotic spindle where it phosphorylates FAM83D (and presumably other substrates) to regulate spindle alignment. The experiments are in general rigorous and done using state of the art elegant methods. The text is clear and easy to read. The figures are well assembled. The conclusions are interesting and provide an important insight into how the ubiquitous CK1 α can do so many things.

A significant limitation of the manuscript is that virtually all studies were performed in a single cell type, U2OS. I would suggest that several key interaction studies be performed in other cell types as well to test if this is generalizable.

Other comments:

Multiple antibodies against several of the proteins were used and it isn't clear which antibody was used in which experiment; e.g. for CK1 α , there are three sources. For example, in Fig EV4, which of these was used in the various immunoblots?

Anti-CK1 α (cat.: A301-991A, 1:1000) was from Bethyl

anti-CK1 α (SA527, 3rd bleed, 1:1000) Dundee

immunofluorescence, anti-CK1 α (cat.: sc-6477, 1:100) was from Santa Cruz

I strongly suggest the antibody section of the methods contain a table of the antibodies used listing the antigen (including epitope when known), source/catalog number, species (rat, mouse, rabbit, alpaca, etc.), dilution, which figure the antibody was used in, and importantly, a reference to demonstrate the validation of the specific lot of antibody. For example, how was this lot of Santa Cruz antibody validated? This may require an additional figure with validation of each antibody, or simply reference to loss of signal in a knockout or shift in molecular weight with addition of a tag (e.g. as in Fig EV4). This may sound picky but especially with a discontinued Santa Cruz antibody from a company with a sordid history of quality control problems, this is important.

Fig 4: HMMR recruits the FAM83D/ CK1 α complex to the spindle. Does CK1 α bind to FAM83D in cells lacking HMMR?

Fig 5B: the rescued mobility shift of FAM83D appears to be less than the mobility shift of wildtype. This should be commented upon. Why might this be?

It would be helpful if the authors would speculate why CK1 α only binds to FAM83D in mitosis. Could this be because of HMMR, or perhaps phosphorylation of FAM83D by a CDK?

Referee #2:

The correct positioning of the mitotic spindle is the key for error-free cell division and the proper development. In most instances, accurate spindle positioning requires either the interaction between the astral microtubules and the cortical dynein and/or robust subcortical actin cytoskeleton. In this manuscript, the authors uncovered that FAM83D (CHICA) specifically interacts with CK1 α during mitosis, where CK1 α localizes prominently enrich at the spindle pole with the help of FAM83D. Also, cells that lack either FAM83D, or FAM83D interaction with CK1 α show a substantial delay in mitotic exit, and also reveal spindle positioning defects. Interestingly, artificial delivering CK1 α in cells expressing a mutant form of FAM83D that is unable to interact with CK1 α partly rescue both mitotic delays as well as spindle positioning.

This manuscript by Fulcher et al., is interesting as the authors have attempted to characterize the function of CK1 α in spindle orientation in mammalian cells. The strong point of this work is the discovery of a new kinase (CK1 α) in spindle positioning in human cells. The weakness is that no attempt has been made to study how CK1 α localization at the mitotic spindle influence spindle positioning (molecular mechanism; and see few major points related to that). Therefore, I feel that because of the already existing literature related with the FAM83D (CHICA) on mitotic delay, and spindle positioning, the scope of this study is somewhat limited until authors put substantial efforts in illustrating the mechanism by which CK1 α regulates spindle positioning in mammalian cells.

Major points:

1. Control cells when grown on L-shape micro-patterns cell position their mitotic spindle along the hypotenuse and the authors observed significant spindle orientation defects in cells lacking FAM83D or expressing FAM83D(F283A). Interestingly the impact of FAM83D(F283A) on spindle orientation was rescued in cells where CK1 α was targeted to the spindle with the help of FAM83D(F283A). CK1 α -mediated phosphorylation may regulate spindle orientation in control cells downstream of FAM83D, can optogenetic targeting of CK1 α in FAM83D knock out cells [not in FAM83D(F283A)] rescue spindle orientation as well as a mitotic delay? Also, since at many instances a delay in the mitotic progression (metaphase to anaphase transition) can impact spindle orientation, it may well be that the spindle orientation defect in the absence of CK1 α at mitotic spindle is rather indirect.
2. It was not clear to me what is the impact of CK1 α loss (RNAi or KO) on spindle positioning, mitotic progression and actin cytoskeleton?
3. FAM83D, as well as CK1 α , are enriched at the spindle, how the authors envisage that loss of FAM83D impact actin cytoskeleton, and more importantly, how CK1 α targeting to the spindle can rescue this in cells expressing FAM83D(F283A)?
4. Recently, FAM83D interacting partner HMMR was shown to affect spindle positioning, and it is involved in Plk1-dependent spindle positioning pathways. It was also shown in loss of HMMR affects levels of active Plk1 at the spindle pole, as well astral microtubules (Connell et al., 2017, eLIFE). Therefore, I am wondering what the impact of FAM83D loss or FAM83D(F283A) mutation on active Plk1 levels at the centrosome as well as on the dynamic astral microtubules.

Minor issues:

1. A more thorough discussion is lacking in the manuscript without citing a great amount of literature in the field of spindle positioning. For instance, in *C. elegans* embryo casein kinase 1 (CSNK-1), regulates spindle positioning by regulating cortical force generator (Panbianco et al., 2008), and thus it should be mentioned. Beyond Cdk1, a vast amount of literature has linked the role of major mitotic kinases such as Plk1, Aurora A in spindle positioning (Kiyomitsu and Cheeseman, 2012; Tame et al., 2016; Sana et al., 2018; Gallini et al., 2016; Kotak et al., 2016), and thus it should be appropriately mentioned in the manuscript.
2. The overall levels of CK1 α appear to be dramatically low in the KO of FAM83D (Figure 2A), could it be just a stability issue? If yes, can authors show the levels of CK1 α in FAM83D KO cells which are simultaneously treated with proteasome inhibitor MG132 in metaphase cells by immunostaining?
3. Also, in Figure 4C and 4D can authors should analyze the stability of FAM83D upon mitotic exit in the presence of MG132 (RO-3306+MG132), this would be crucial to address whether proteasome alone or mitotic exit based posttranslational modification are at play for controlling FAM83D stability.
4. Authors show that CK1 α is unable to directly phosphorylate FAM83D, however the CK1 α interaction with the FAM83D is must for the electrophoretic mobility shift, and authors suggests that priming phosphorylation may be required for FAM83D phosphorylation by CK1 α . Since several mitotic kinases localize to the spindle, and several specific inhibitors are available to test the

function of such mitotic kinases, it would be straightforward to test which kinase-mediating priming is crucial for such mobility shift.

1st Revision - authors' response

29 April 2019

Referee #1:

Fulcher and coworkers identify a mitosis-specific interaction between FAM83D and CK1 α that recruits CK1 α to the mitotic spindle where it phosphorylates FAM83D (and presumably other substrates) to regulate spindle alignment. The experiments are in general rigorous and done using state of the art elegant methods. The text is clear and easy to read. The figures are well assembled. The conclusions are interesting and provide an important insight into how the ubiquitous CK1 α can do so many things.

Response: We thank the referee for the critical and constructive appraisal of our manuscript.

A significant limitation of the manuscript is that virtually all studies were performed in a single cell type, U2OS. I would suggest that several key interaction studies be performed in other cell types as well to test if this is generalizable.

Response: We thank the referee for highlighting the limitation. In the original manuscript, in addition to the U2OS cell data, we also showed that CK1 α localises to the mitotic spindle in mouse embryonic fibroblasts derived from wild type but not *HMMR* knockout mice (Figure 4I). We have now expanded endogenous FAM8D:CK1 α interaction assays to three additional human cancer cell lines – HeLa, A549 and HaCaT – and show in all cases that FAM83D is phosphorylated (as shown by the mobility shift) in mitosis, and interacts with CK1 α in mitosis. We have added this figure (**new Figure 1K**) to the manuscript, and insert it below for your perusal:

Figure 1K: Asynchronous (AS) or nocodazole-synchronised mitotic (M) WT HeLa, A549 and HaCaT cells were subjected to immunoprecipitation (IP) with either IgG- or anti-FAM83D-coupled sepharose beads. Input and IP samples were analysed by immunoblotting (IB) with the indicated antibodies.

Other comments:

Multiple antibodies against several of the proteins were used and it isn't clear which antibody was used in which experiment; e.g. for CK1 α , there are three sources. For example, in Fig EV4, which of these was used in the various immunoblots?

Anti-CK1 α (cat.: A301-991A, 1:1000) was from Bethyl

anti-CK1 α (SA527, 3rd bleed, 1:1000) Dundee

immunofluorescence, anti-CK1 α (cat.: sc-6477, 1:100) was from Santa Cruz

I strongly suggest the antibody section of the methods contain a table of the antibodies used listing the antigen (including epitope when known), source/catalog number, species (rat, mouse, rabbit, alpaca, etc.), dilution, which figure the antibody was used in, and importantly, a reference to demonstrate the validation of the specific lot of antibody. For example, how was this lot of Santa Cruz antibody validated? This may require an additional figure with validation of each antibody, or simply reference to loss of signal in a knockout or shift in molecular weight with addition of a tag (e.g. as in Fig EV4). This may sound picky but especially with a discontinued Santa Cruz antibody from a company with a sordid history of

quality control problems, this is important.

Response: We agree with the reviewer that this is important to clarify. As such, we have updated the **Methods** section to include a table describing all the antibodies used (**new Table 1**). The table also includes references for antibody specificity, and details which figures each antibody was used in.

Fig 4: HMMR recruits the FAM83D/ CK1 α complex to the spindle. Does CK1 α bind to FAM83D in cells lacking HMMR?

Response: We thank the reviewer for this suggestion and we spent a significant time during revisions to address this. However, we encountered several technical challenges that have prevented us from providing a definitive answer. First, the anti-FAM83D and anti-CK1 α antibodies which were raised against the respective human protein antigens failed to effectively immunoprecipitate endogenous FAM83D and CK1 α from wild type and *HMMR*^{-/-} mouse embryonic fibroblasts. Second, these immortalised MEFs proved extremely resistant to transient transfections of plasmids encoding human FAM83D and CK1 α . Third, the rates of synchronisation in these cells after STLC treatment were extremely low (possibly because of the transformation process), which translated into low yields of mitotic protein extracts, and subsequently, the typical protein concentration we would use for IPs from mitotic extracts was extremely challenging to achieve. Currently we do not have any *HMMR* knockout human cell line, which would mitigate these issues. Although we are in the process of generating such a cell line with CRISPR/Cas9, this has been challenging as well and as yet we do not have any *HMMR* knockout clones.

However, two sets of observations suggest that the FAM83D and CK1 α interactions are not likely to occur in *HMMR* knockout cells: i. the FAM83D mitotic mobility shift evident in WT MEFs does not occur in *HMMR* knockout MEFs; and ii. the mitotic mobility shift of FAM83DGFP in *FAM83DGFP/GFP* knockin U2OS cells is also evident in asynchronous conditions when FAM83D-GFP is targeted with anti-GFP nanobody (aGFP.16) bound CK1 α (**new Figure EV5A**; see below in response to the next comment), suggesting that the interaction between FAM83D and CK1 α is essential to cause the mobility shift in FAM83D.

In any case, whilst we cannot definitively say if FAM83D and CK1 α still associate or not in the absence of *HMMR*, we can say that CK1 α cannot localise to the spindle if *HMMR* or FAM83D are not present. The spindle recruitment of endogenous CK1 α through FAM83D and *HMMR* is the main point of the set of experiments we have included in the manuscript.

Fig 5B: the rescued mobility shift of FAM83D appears to be less than the mobility shift of wildtype. This should be commented upon. Why might this be?

Response: We thank the reviewer for bringing this to our attention. Firstly, to rule out the possibility that aGFP.16-CK1 α is phosphorylating FAM83D(F283A)-GFP on random sites, we artificially recruited aGFP.16-CK1 α to the wild-type FAM83D-GFP knockin protein via retroviral infection, and compared the mobility shift between these cells and uninfected controls. The mitotic mobility-shift is the same in both the infected and non-infected cells, suggesting that the phosphorylation of FAM83D imparted by aGFP.16-CK1 α is not random, and that, most likely, the same sites are phosphorylated by endogenous CK1 α and aGFP.16-CK1 α .

Thus, we speculate that the CK1-binding-deficient FAM83D(F283A) mutant does not display the same mobility shift as the wild-type FAM83D due to a potential structural change imparted by the F283A mutation. This F283A mutation may interfere with CK1 α (or other downstream kinases) accessing some of the phospho-residues, and thus the full phosphorylation status is not achieved.

A new figure with the WT aGFP.16-CK1 α is included as an extended view figure (**new Fig. EV5A**) and we provide it below for your perusal. We have expanded the results section to reflect these changes and the interpretation.

Figure EV5A: STLC-synchronised mitotic (M) *FAM83D*^{GFP/GFP} knockin cells and *FAM83D*^{GFP/GFP} knockin cells stably expressing aGFP.16-CK1α were subjected to GFPTRAP immunoprecipitation (IP), followed by immunoblotting (IB) with the indicated antibodies. Asynchronous (AS) cells were used as controls.

It would be helpful if the authors would speculate why CK1α only binds to FAM83D in mitosis. Could this be because of HMMR, or perhaps phosphorylation of FAM83D by a CDK?

Response: We agree with the reviewer that this is an interesting and outstanding question.

Rather than just speculate, we set out to try and address this issue experimentally! When we express the FAM83D N-terminus (DUF1669) alone, we observe a robust interaction with CK1α. However, the C-terminus of FAM83D alone, which lacks the DUF1669, does not pull down CK1α. When we co-express the N-terminal and C-terminal fragments and pull down the N-terminus, we observe C-terminal fragments coimmunoprecipitating, suggesting that both termini associate with each other. In this case, due to excess N-terminal fragments, we can still detect CK1α in the N-terminal fragment IPs. However, when we co-express both the N- and C-terminal fragments and pull down the C-terminal fragment, we again observe robust association with the N-terminal fragment yet, in this case, no CK1α coimmunoprecipitates. Collectively these data suggest that the C-terminus of FAM83D inhibits the N-terminus from binding to CK1α.

Thus, we think that during mitosis FAM83D is modified in a way that relieves the inhibitory role of the C-terminus on the N-terminus, allowing the N-terminus access to CK1α binding. The underlying mechanisms for this could be either a post-translational modification (e.g. phosphorylation or dephosphorylation), or it could be due to mechanical forces when FAM83D associates with HMMR (and/or other proteins such as DYNLL1) on the spindle. We have explored the effects of mutating some of the identified mitotic FAM83D phosphosites to alanine, but these phospho-mutants, either alone or in combination, did not seem to impact the CK1α association. Thus, the exact mechanism regulating this remains an elusive, and our future endeavours will attempt to resolve this very interesting puzzle.

The new data are included as new panels in **new Figs. EV2C&D**, and are included below for your perusal:

Figs. EV2C&D: *FAM83D*^{-/-} U2OS cells were transiently transfected with plasmids encoding full-length GFP-FAM83D (FL), isolated GFP-tagged FAM83D N-terminus (N), isolated FLAG-tagged FAM83D C-terminus (C), or with both N and C fragments together. Cells were lysed and subjected to GFP-TRAP immunoprecipitations (IP). Whole cell lysate (input) and IP samples were separated by SDS-PAGE, before immunoblotting with the indicated

antibodies. **D:** As in C. except that the FL plasmid was omitted, and FLAG IPs were performed instead of GFP IPs.

Referee #2:

The correct positioning of the mitotic spindle is the key for error-free cell division and the proper development. In most instances, accurate spindle positioning requires either the interaction between the astral microtubules and the cortical dynein and/or robust subcortical actin cytoskeleton. In this manuscript, the authors uncovered that FAM83D (CHICA) specifically interacts with CK1 α during mitosis, where CK1 α localizes prominently enrich at the spindle pole with the help of FAM83D. Also, cells that lack either FAM83D, or FAM83D interaction with CK1 α show a substantial delay in mitotic exit, and also reveal spindle positioning defects. Interestingly, artificial delivering CK1 α in cells expressing a mutant form of FAM83D that is unable to interact with CK1 α partly rescue both mitotic delays as well as spindle positioning.

This manuscript by Fulcher et al., is interesting as the authors have attempted to characterize the function of CK1 α in spindle orientation in mammalian cells. The strong point of this work is the discovery of a new kinase (CK1 α) in spindle positioning in human cells. The weakness is that no attempt has been made to study how CK1 α localization at the mitotic spindle influence spindle positioning (molecular mechanism; and see few major points related to that). Therefore, I feel that because of the already existing literature related with the FAM83D (CHICA) on mitotic delay, and spindle positioning, the scope of this study is somewhat limited until authors put substantial efforts in illustrating the mechanism by which CK1 α regulates spindle positioning in mammalian cells.

Response: We thank the reviewer for an in depth and critical review of our manuscript. We would like to respectfully emphasize that our novel findings include:

- FAM83D interacts and co-localises with CK1 α at the mitotic spindle during mitosis, all demonstrated at the endogenous level.
- *FAM83D*^{-/-} cells or those that harbour CK1 α -binding deficient *FAM83DF283A/F283A* knockin mutation no longer recruit CK1 α to the spindle, and display pronounced spindle positioning defects, and a delayed metaphase-to-anaphase transition.
- We rescue these defects in *FAM83D*^{-/-} cells by either restoring *FAM83D* at the endogenous locus using a novel CRISPR/Cas9 knockin strategy, or by using nanobody-directed recruitment to deliver wild-type CK1 α to the spindle in *FAM83DF283A/F283A* knockin cells.
- Importantly, nanobody-directed recruitment of *catalytically-inactive CK1 α* in *FAM83DF283A/F283A* knockin cells did not restore these defects.
- • HMMR directs FAM83D to the mitotic spindle and so, in *HMMR*^{-/-} MEFs, both FAM83D and CK1 α fail to localise to the mitotic spindle.

Of course, identifying key CK1 α substrates at the mitotic spindle that define proper spindle orientation is desirable, but clearly beyond the scope of the current manuscript. For most mitotic kinases, even for some of the earliest ones to be identified decades ago, the mitotic substrate landscape is still not fully elucidated, and often, identification and validation of a single substrate for any mitotic kinase prompts publication in reputed journals. Our current work **unequivocally places CK1 α activity on the centrosomes and mitotic spindles, provides molecular mechanisms through which this happens, and highlights the spindle-misorientation and mitotic consequences of disrupting CK1 α delivery to the spindle.** Identifying FAM83D-dependent CK1 α substrates in mitosis and how these ensure proper and timely mitotic progression is our next goal and we have included this in the Discussion.

On another note, CK1 α and other CK1 isoforms have long been deemed undruggable due to their participation in multiple, diverse biological processes. Thus, the work presented here, in agreement with our published data on other FAM83 proteins, suggests that targeting individual FAM83:CK1 interactions might be an effective means of inhibiting CK1 in distinct physiological processes. Of relevance here, inhibiting the FAM83D-CK1 α interaction would effectively block CK1 α from localising to the spindle, whilst having minimal effect on the

other CK1 α -relevant cellular functions.

We have also incorporated additional mechanistic information into the manuscript which provides further insight into how the FAM83D-CK1 α interaction is regulated in cells (see response to reviewer 1).

Major points:

*1. Control cells when grown on L-shape micro-patterns cell position their mitotic spindle along the hypotenuse and the authors observed significant spindle orientation defects in cells lacking FAM83D or expressing FAM83D(F283A). Interestingly the impact of FAM83D(F283A) on spindle orientation was rescued in cells where CK1 α was targeted to the spindle with the help of FAM83D(F283A). CK1 α -mediated phosphorylation may regulate spindle orientation in control cells downstream of FAM83D, **can optogenetic targeting of CK1 α in FAM83D knock out cells [not in FAM83D(F283A)] rescue spindle orientation as well as a mitotic delay?** Also, since at many instances a delay in the mitotic progression (metaphase to anaphase transition) can impact spindle orientation, it may well be that the spindle orientation defect in the absence of CK1 α at mitotic spindle is rather indirect.*

Response: We thank the reviewer for this suggestion. First, the best way to rescue defects in *FAM83D* knockout cells is by restoring wild-type *FAM83D* into these cells. Not only did we do this, but we ensured that the rescue was done at the cell-cycle-regulated native *FAM83D* locus by a novel CRISPR/Cas9 strategy. **We believe this sets a new precedent on how rescue experiments should be performed for all future knockout studies.** Second, as the referee correctly points out, we were able to identify a CK1 α -interaction deficient mutant of FAM83D (i.e. a single amino acid, F283A, substitution), knock it in, again at the native *FAM83D* locus (albeit with a GFP-tag) with CRISPR/Cas9, and show that this mutant was defective in recruiting CK1 α to the mitotic spindle and driving timely mitosis. Importantly, this mutant still localised to the mitotic spindle and by mass-spectrometry pulled down every other interactor of wild-type FAM83D except CK1 α . It is in this context that we show that artificially taking aGFP.16-CK1 α to the mitotic spindle on the FAM83D(F283A)-GFP mutant rescues the defects. Taking CK1 α artificially to the mitotic spindle outside of the FAM83D context in *FAM83D* knockout cells would be potentially problematic in that seven other FAM83 proteins (and potentially their associated partners) that also interact with CK1 α would be predicted to interfere with CK1 α at the spindle, and might result in artefacts that are extremely hard to interpret. Therefore, we feel the optogenetic delivery of CK1 α in FAM83D-knockout cells, while an attractive proposition, might produce data that are hard to interpret as it lacks the FAM83D context. Furthermore, we lacked the expertise and infrastructure to perform the suggested optogenetic targeting experiments within the given time frame and, as explained already, we believe we have done the rescue experiments in the best way possible.

2. It was not clear to me what is the impact of CK1 α loss (RNAi or KO) on spindle positioning, mitotic progression and actin cytoskeleton?

Response: We thank the reviewer for bringing this to our attention. We were unable to generate CK1 α -knockout cells by CRISPR/Cas9 (no homozygous knockout clones identified), and in line with this, a CRISPR study has previously identified *CSNK1A1* (CK1 α encoding gene) as an essential gene (Wang, Birsoy et al., 2015).

Therefore, we transiently depleted CK1 α (~70% depletion compared to control cells) in U2OS cells using siRNAs, and under these conditions, we observed robust spindle positioning defects, as well as a delay in the metaphase-to-anaphase transition. We include these data in **new Appendix Figure S1** and include them below for your perusal.

However, these data have to be taken cautiously as CK1 α also interacts with seven other FAM83 members and has functions in multiple, diverse signalling processes beyond just mitosis. Therefore, the mitotic phenotypes observed with CK1 α siRNAs, despite phenocopying FAM83D KO cells, could also be due to indirect consequence(s) of other CK1 α function being affected. This is precisely why we believe that targeting regulatory subunits of CK1 α (such as FAM83 proteins) present great promise for shutting down specific CK1 α -dependent processes, and this is what we see as a big strength of the present study.

Appendix Figure S1: A: Western blot analysis of U2OS cell extracts 48 h after treatment with scrambled siRNA (siScramble) controls or siRNA targeting CK1a (siCK1a). **B:** Representative images of mitotic U2OS cells stained with Hoechst, beginning at metaphase and taken every 5 min as they progressed through division. Mitotic stage was determined by chromosome condensation and is indicated by the coloured boxes. Scale bar 20 μ m. **C:** Graphical representation of the kinetics of transition from metaphase alignment (yellow) to anaphase (green), and cytokinesis (blue), determined by the morphology of chromosomes and daughter cells, respectively. The kinetics for 100 mitotic cells per genotype are plotted as measured for 50 cells per experiment from 2 independent experiments. **D:** Length of time needed to transition from metaphase to anaphase. Mean \pm SEM is plotted for 2 independent experiments, which each measured 50 mitotic cells per genotype (n= 100 mitotic cells per genotype total). *** p< 0.0001, Student's *T*-test. **E:** Representative images of mitotic U2OS cells stained with Hoechst and grown on Lshaped micropatterns previously coated with fibronectin. The cell division angle at anaphase is indicated (yellow line). Scale bar = 20 μ m. **F:** Circular graphs, superimposed on a L-shaped micropattern, show the distribution of cell division angles measured at anaphase. Angles are plotted for 100 U2OS cells per genotype measured from 2 independent experiments. The percentages of division angles $>$ 15 $^\circ$ from the expected axis (red line) are indicated. **G:** Heatmap additive intensities of RFP-actin localization in two representative mitotic U2OS cells for each genotype grown on fibronectin-coated, L-shaped micropatterns. Arrowheads indicate polarized cortical actin.

3. *FAM83D*, as well as *CK1a*, are enriched at the spindle, how the authors envisage that loss of *FAM83D* impact actin cytoskeleton, and more importantly, how *CK1a* targeting to the spindle can rescue this in cells expressing *FAM83D(F283A)*?

Response: We thank the reviewer for raising this point. We think a likely explanation is that phosphorylation of specific substrates by CK1a is a key event in efficient mammalian cell division. In the absence of their phosphorylation, the spindle is unable to be effectively orientated. Currently, whether these substrates are localised on the mitotic spindle itself, or the subcortical actin clouds that also regulate spindle positioning, is unclear. **We have added a sentence to this effect in the discussion** to address this point, as we believe it is important to mention.

4. Recently, *FAM83D* interacting partner *HMMR* was shown to affect spindle positioning, and it is involved in *Plk1*-dependent spindle positioning pathways. It was also shown in loss of *HMMR* affects levels of active *Plk1* at the spindle pole, as well astral microtubules (Connell *et al.*, 2017, *eLIFE*). Therefore, I am wondering what the impact of *FAM83D* loss or *FAM83D(F283A)* mutation on active *Plk1* levels at the centrosome as well as on the dynamic astral microtubules.

Response: We thank the reviewer for this excellent suggestion. We were also considering this possibility, and indeed it was the Connell *et al* paper that prompted us to initiate collaboration with the Maxwell lab for the present study (Chris Maxwell, a co-author on this manuscript, is the corresponding author on the Connell *et al* paper). We had focussed more on the spindle positioning assays, but we performed the suggested experiments to see if *FAM83D* also acts in the *PLK1*-dependent spindle positioning pathway, like *HMMR*. However, it should be noted that *HMMR* is also involved in recruiting the *TPX2*-AuroraA kinase complex to the mitotic spindle. Thus, removing *HMMR* from cells would be predicted to have a more profound impact on spindle positioning and mitosis, than removing *FAM83D* (or *TPX2*) alone would.

That said, when CK1a is not able to localise to the spindle, we observe a reduction in active PLK1 at spindle poles and kinetochores, suggesting that CK1a may act in the PLK1-dependent pathway. One likely explanation is that CK1a phosphorylates PLK1-docking proteins on the spindle pole to enable polo-box domain binding to the phosphoprotein, and hence PLK1 localisation. Alternatively, CK1a may be involved in the activation of PLK1 at spindle poles. However, in our FAM83D pulldowns, we did not detect any PLK1 by mass spectrometry, suggesting that any effects of FAM83D-bound CK1a on PLK1 are likely to be either indirect or involve transient interaction(s).

While these data are hugely promising, we believe they raise interesting questions on the mechanisms involved and clearly require a substantial body of further work to fully ascertain whether CK1a directly acts in the PLK1-dependent pathway, and where within the pathway CK1a acts. We will be following up on these observations in the future. We include these new data below for your perusal in rebuttal Figure 1.

Rebuttal Figure 1: **A.** Localization of phospho-PLK1 (p-PLK1 (Thr210)) at spindle poles (TUBG1) in U2OS cells 48 hours after treatment with scrambled siRNA (siScramble) or siRNA targeting CK1a (siCK1a) as well as in FAM38D F283A cells and FAM38D F283A + CK1a cells. Scale bars 10 μ m. **B.** Localization of p-PLK1 at kinetochores (BubR1) in U2OS cells 48 hours after treatment with siScramble or siCK1a as well as in FAM38D F283A cells and FAM38D F283A + CK1a cells. Dashed line indicates line graph shown in panel C. Scale bars 10 μ m. **C.** Line graphs of CK1a and p-PLK1 intensity from pole to pole in U2OS cells with indicated genotype or siRNA treatment. **D.** Quantification of p-PLK1 intensity at spindle pole (TUBG1) and kinetochores (BubR1) in U2OS cells with indicated genotype or siRNA treatment. Mean \pm SEM are plotted for two independent experiments, which each measured 15 mitotic cells per treatment. Individual cell measurements (n= 30 per treatment) are also shown. *** p< 0.001, ANOVA. **E.** Astral microtubules contact the cortex in U2OS cells with indicated genotype or siRNA treatment. Yellow box indicates EB1 inset and white dotted line indicates region of cell cortex. Scale bars 10 μ m. **F.** Quantification of EB1 at the cortex in U2OS cells with indicated genotype or siRNA treatment. Mean \pm SEM are plotted for two independent experiments, which each measured 20 mitotic cells per treatment. *** p< 0.005, ANOVA.

Minor issues:

1. A more thorough discussion is lacking in the manuscript without citing a great amount of literature in the field of spindle positioning. For instance, in *C. elegans* embryo casein kinase 1 (CSNK-1), regulates spindle positioning by regulating cortical force generator (Panbianco et al., 2008), and thus it should be mentioned. Beyond Cdk1, a vast amount of literature has linked the role of major mitotic kinases such as Plk1, Aurora A in spindle positioning (Kiyomitsu and Cheeseman, 2012; Tame et al., 2016; Sana et al., 2018; Gallini et al., 2016; Kotak et al., 2016), and thus it should be appropriately mentioned in the manuscript.

Response: We thank the referee for providing references to these fantastic studies, and have updated the **Discussion** section to include them.

2. The overall levels of CK1a appear to be dramatically low in the KO of FAM83D (Figure 2A), could it be just a stability issue? If yes, can authors show the levels of CK1a in FAM83D

KO cells which are simultaneously treated with proteasome inhibitor MG132 in metaphase cells by immunostaining?

Response: We thank the referee for raising this query. The spindle is a very bright and prominent structure when viewed by immunofluorescence. Because of that, the optimal exposure of the images in Figure 2A was based on the spindle being visible and not saturated. When that happens, the cytoplasmic CK1 α staining behind and surrounding the spindle appears faint. However, if one were to remove the spindle and increase the exposure, we would see that the cytoplasmic levels of CK1 α are the same between WT and FAM83D KO cells. Additionally, in all of our immunoblotting figures, we do not observe any substantial decrease in CK1 α protein levels between mitotic and asynchronous cells.

3. Also, in Figure 4C and 4D can authors should analyze the stability of FAM83D upon mitotic exit in the presence of MG132 (RO-3306+MG132), this would be crucial to address whether proteasome alone or mitotic exit based posttranslational modification are at play for controlling FAM83D stability.

Response: We thank the referee for this excellent suggestion. We have now performed the suggested experiment and observe robust FAM83D degradation following forced mitotic exit with RO-3306. This degradation is halted in the presence of MG132. Thus, it appears that FAM83D is rapidly degraded via the proteasome, following mitotic exit. This data is included as a new figure (**new Fig 4E**) and is included below for your perusal:

Fig 4E: Wild-type U2OS cells were either left asynchronous (AS), or arrested in mitosis with STLTC and collected by shake-off (M). AS and M cells were incubated in media containing combinations of RO-3306 and MG132 as indicated, prior to lysis. MG132 was applied for 1.5 h, whereas RO-3306 was applied for the last 1 h of incubation. Samples were lysed and subjected to SDS-PAGE, before immunoblotting (IB) with the indicated antibodies.

4. Authors show that CK1 α is unable to directly phosphorylate FAM83D, however the CK1 α interaction with the FAM83D is must for the electrophoretic mobility shift, and authors suggests that priming phosphorylation may be required for FAM83D phosphorylation by CK1 α . Since several mitotic kinases localize to the spindle, and several specific inhibitors are available to test the function of such mitotic kinases, it would be straightforward to test which kinase-mediating priming is crucial for such mobility shift.

Response: We thank the referee for highlighting this point. We have performed the suggested experiments, and have not been able to induce FAM83D phosphorylation by the main mitotic kinase CDK1/Cyclin B, PLK1, Aurora A or NEK6/7. The catalytic activities of these kinases were verified and measured against their respective peptide substrates by the MRC kinase profiling service (Dundee). Furthermore, their inhibitors in cells have not been effective at inhibiting FAM83D phosphorylation-induced mobility shift, with the exception of CDK1/Cyclin B inhibition once cells are already in mitosis, but this treatment causes cells to exit mitosis, and thus is likely an indirect effect resulting from the forced mitotic exit. Thus, this remains an outstanding, yet interesting question, that will be teased out during future

endeavours. We include the kinase assay data below for your perusal in rebuttal Figures 2 & 3.

Rebuttal Figure 2: An in vitro kinase assay was set up with recombinant FAM83D and the indicated kinases. Following 30 min incubation with radioactive ATP at 30°C, samples were denatured and separated by SDS-PAGE. The gel was stained with Coomassie blue, and exposed to x-ray films for 16 h overnight. FAM83G was included as a positive control for CK1α activity. The other kinases included in the screen were profiled by the Division of Signal Transduction Therapy (University of Dundee), and were used under conditions where they are catalytically active.

Rebuttal Figure 3: An in vitro kinase assay was set up with recombinant FAM83D and the indicated kinases. Following 30 min incubation with radioactive ATP at 30°C, samples were denatured and separated by SDS-PAGE. The gel was stained with Coomassie blue, and exposed to x-ray films for 16 h overnight. FAM83G was included as a positive control for CK1 α activity, and Histone H1 was included as a positive control for CDK1/Cyclin B activity. The other kinases included in the screen were profiled by the Division of Signal Transduction Therapy (University of Dundee), and were used under conditions where they are catalytically active.

References:

Wang T, Birsoy K, Hughes NW, Krupczak KM, Post Y, Wei JJ, Lander ES, Sabatini DM (2015) Identification and characterization of essential genes in the human genome. *Science* 350: 1096-101.

2nd Editorial Decision

11 June 2019

Thank you for submitting the revised version of your manuscript. It has now been seen by two of the original referees.

As you can see, both referees find that the study is significantly improved during revision and recommend publication. Before I can accept the manuscript, I need you to address the below minor/editorial points:

- Please address the remaining concerns of referee #2. Regarding point 1, I can see that you looked at actin cortex morphology in Appendix Figure S1 under CK1 α kd conditions. As for the point 2, since this experiment was not recommended in the previous review, I don't deem necessary to address this point experimentally for publication. However, please respond to both points textually.

Thank you again for giving us to consider your manuscript for EMBO Reports, I look forward to your revision.

REFeree REPORTS

Referee #1:

The authors have addressed the issues raised. This is an important addition to our knowledge of CK1 and mitosis.

Referee #2:

Authors have substantially improved the revised version of the manuscript and have adequately addressed my previous concerns. I have now only a few remaining concerns:

1. Appendix Figure S1 revealing the siRNAs mediated depletion of siRNA CK1 α must be included in the manuscript. Earlier, I also suggested the authors to analyse the impact of CK1 α (RNAi) on the actin cytoskeleton; I guess that has not been addressed in the Appendix Figure S1. I like to know if the spindle orientation defects observed upon CK1 α depletion stem from its impact on the actin cortex?
2. Also, if CK1 α -mediated phosphorylation impact mobility of FAM83D in mitosis, I like to see what happen to the FAM83D phosphorylation-induced mobility shift in mitotically synchronised extracts made from CK1 α (RNAi) cells?

Referee #1:

The authors have addressed the issues raised. This is an important addition to our knowledge or CK1 and mitosis.

Response: We thank the referee for their critical appraisal of our manuscript and appreciate all the feedback.

Referee #2:

Authors have substantially improved the revised version of the manuscript and have adequately addressed my previous concerns. I have now only a few remaining concerns:

Response: We thank the reviewer for providing a thorough and critical review of our manuscript, and for suggestions and feedback throughout the review process.

1. Appendix Figure S1 revealing the siRNAs mediated depletion of siRNA CK1 α must be included in the manuscript. Earlier, I also suggested the authors to analyse the impact of CK1 α (RNAi) on the actin cytoskeleton; I guess that has not been addressed in the Appendix Figure S1. I like to know if the spindle orientation defects observed upon CK1 α depletion stem from its impact on the actin cortex?

Response: The Appendix S1 figure is **called out** in the main text and will be published alongside the manuscript. With a limit on EV figures to only 5, we chose to include this and one other figure as Appendix Figures. One consideration for this, as we pointed out in our responses after the first round of reviews, is because the mitotic phenotypes observed with CK1 α siRNAs, despite phenocopying *FAM83D* knockout cells, could also be due to indirect consequence(s) of other CK1 α functions being affected, as siRNA knockdown of CK1 α potentially impacts all eight FAM83-CK1 α complexes. Putting these data in the main figures might give an impression that siRNA knockdown of CK1 α only affects the FAM83D-CK1 α complex in mitosis, which we do not think is the case.

Regarding the impact of CK1 α knockdown on the actin cytoskeleton, as the editor has pointed out, we did perform the suggested experiment and it was included in **Figure S1G** and **called out** in the main text. Cells with siRNA-mediated knockdown of CK1 α follow the same pattern as *FAM83D* knockout cells in terms of their actin cytoskeletal organisation.

2. Also, if CK1 α -mediated phosphorylation impact mobility of FAM83D in mitosis, I like to see what happen to the FAM83D phosphorylation-induced mobility shift in mitotically synchronised extracts made from CK1 α (RNAi) cells?

Response: We thank the reviewer for this suggestion and whilst it would be interesting to generate this data, this would not add to the core hypothesis that FAM83D delivers CK1 α to the mitotic spindle for proper spindle positioning. In line with the editor's recommendation, we speculate that siRNA against CK1 α , which causes a substantial but not complete depletion in levels of CK1 α protein, would cause at least a partial collapse of the mitotic FAM83D mitotic mobility shift.

Thank you for submitting your revised manuscript. I have now taken a look at everything and all is fine. Therefore I am very pleased to accept your manuscript for publication in EMBO Reports.

Corresponding Author Name: Gopal Sapkota

Manuscript Number: EMBOR-2018-47495